# Metabolic and Other Endocrine Elements with Regard to Lifestyle Choices: Focus on E-Cigarettes

**DOI:** 10.3390/metabo13121192

**Published:** 2023-12-08

**Authors:** Andrei Osman, Gabriel Sebastian Petrescu, Mihaela Jana Tuculină, Ionela Teodora Dascălu, Cristina Popescu, Anca-Ștefania Enescu, Constantin Dăguci, Anca-Pati Cucu, Claudiu Nistor, Mara Carsote

**Affiliations:** 1Department of Anatomy and Embryology, Faculty of Dental Medicine, University of Medicine and Pharmacy of Craiova, Department ENT & Clinical Emergency County Hospital of Craiova, 200349 Craiova, Romania; andreiosman3@gmail.com (A.O.); cristina.popescu25@yahoo.com (C.P.); anca.stefania.enescu@gmail.com (A.-Ș.E.); 2Department of Oral and Maxillofacial Surgery, Faculty of Dental Medicine, University of Medicine and Pharmacy of Craiova, 200349 Craiova, Romania; sebigabriel_petrescu@yahoo.com; 3Department of Endodontics, Faculty of Dental Medicine, University of Medicine and Pharmacy of Craiova, 200349 Craiova, Romania; mtuculina@yahoo.com; 4Department of Orthodontics, Faculty of Dental Medicine, University of Medicine and Pharmacy of Craiova, 200349 Craiova, Romania; marceldascalu@yahoo.com; 5Department of Oro-Dental Prevention, Faculty of Dental Medicine, University of Medicine and Pharmacy of Craiova, 200349 Craiova, Romania; dagucicristi@yahoo.com; 6PhD Doctoral School, Carol Davila University of Medicine and Pharmacy, Bucharest & Thoracic Surgery Department, Dr. Carol Davila Central Military Emergency University Hospital, 010825 Bucharest, Romania; ancutapati@gmail.com; 7Department 4—Cardio-Thoracic Pathology, Thoracic Surgery II Discipline, Carol Davila University of Medicine and Pharmacy & Thoracic Surgery Department, Dr. Carol Davila Central Military Emergency University Hospital, 010825 Bucharest, Romania; ncd58@yahoo.com; 8Department of Endocrinology, Carol Davila University of Medicine and Pharmacy, Bucharest, Romania & C.I. Parhon National Institute of Endocrinology, 011863 Bucharest, Romania; carsote_m@hotmail.com

**Keywords:** lifestyle, metabolic, endocrine disorder, diabetes, obesity, metabolic syndrome, e-cigarettes, lung, vaping, steroid

## Abstract

Our objective was to overview recent data on metabolic/endocrine disorders with respect to e-cigarette (e-cig) use. This is a narrative review; we researched English, full-length, original articles on PubMed (between January 2020 and August 2023) by using different keywords in the area of metabolic/endocrine issues. We only included original clinical studies (n = 22) and excluded case reports and experimental studies. 3 studies (N1 = 22,385; N2 = 600,046; N3 = 5101) addressed prediabetes risk; N1 showed a 1.57-fold increased risk of dual *vs.* never smokers, a higher risk that was not confirmed in N2 (based on self-reported assessments). Current non-smokers (N1) who were dual smokers still have an increased odd of prediabetes (a 1.27-fold risk increase). N3 and another 2 studies addressed type 2 diabetes mellitus (DM): a lower prevalence of DM among dual users (3.3%) *vs.* cigarette smoking (5.9%) was identified. 6 studies investigated obesity profile (4 of them found positive correlations with e-cig use). One study (N4 = 373,781) showed that e-cig use was associated with obesity in the general population (OR = 1.6, 95%CI: 1.3–2.1, p < 0.05); another (N5 = 7505, 0.82% were e-cig-only) showed that obesity had a higher prevalence in dual smokers (51%) *vs.* cig-only (41.2%, p < 0.05), while another (N6 = 3055) found that female (not male) e-cig smokers had higher body mass index *vs.* non-smokers. Data on metabolic syndrome (MS) are provided for dual smokers (n = 2): one case–control study found that female dual smokers had higher odds of MS than non-smokers. The need for awareness with respect to potential e-cig-–associated medical issues should be part of modern medicine, including daily anamnesis. Whether the metabolic/endocrine frame is part of the general picture is yet to be determined. Surveillance protocols should help clinicians to easily access the medical background of one subject, including this specific matter of e-cig with/without conventional cigarettes smoking and other habits/lifestyle elements, especially when taking into consideration metabolism anomalies.

## 1. Introduction

Healthy lifestyle means not only adequate diet and physical exercise, but also avoiding smoking of conventional cigarettes, and potentially electronic cigarettes (e-cigarettes or e-cig). E-cig, after their launch almost a decade ago, seemed like a useful alternative to conventional smoking, but, further on, various medical issues have been connected with their use, yet, underlying different levels of statistical evidence so far that pointed various side effects and not being completely harmless [1,2,3]. Whether metabolic/endocrine disorders display a particular frame in this particular matter is still an open issue, but metabolic interferences, particularly in patients with increased body mass index (BMI) were identified [1,3].

Placing e-cigarettes among the social and medical reality is mandatory nowadays from a multidisciplinary perspective and a potential supplementary burden. Conventional cigarette smoking, with nicotine burning, is responsible for a multitude of systemic negative effects, requiring the identification of effective means of smoking cessation. So far, no such universally valid method has been identified to help quit smoking, and the introduction of electronic nicotine delivery systems (ENDS) has been regarded as a safe option. However, after more than ten years of their official existence, e-cigarettes and vaping products have revealed unexpected negative effects on health, particularly with cardiopulmonary impact and potentially on metabolic profile. Moreover, the temptation they have represented for children and teenagers has led to an increasing rate of ENDS use among younger people in certain countries [4,5,6,7].

Concerning the general health issues, there are still many areas of debate, and the rapidly changing dynamics of these potential issues are recognized. On the other hand, the recent COVID-19 pandemic brought in prime time many previously known and unknown clinical entities, while syndromes such as EVALI (e-cigarette and vaping-associated lung injury) or VAPI (vaping-associated pulmonary illness) might mimic or interfere with a severe coronavirus infection. Also, the potential increase in e-cigarettes use amid pandemic should be taken into consideration when analyzing new medical data [4,8]. Moreover, the use of e-cig in patients with high cardiovascular risk who required smoking cessation recently increased [9,10,11,12].

### Aim

Our objective was to overview recent scientific data regarding potential metabolic/endocrine-disorders-related frame with respect to e-cigarette use. 

## 2. Methods

This is a narrative review. We researched English, full-length original articles on PubMed between January 2020 and August 2023. The rationale of choosing this specific time frame is related to the recent COVID-19 pandemic and associated regulations such as isolation, outdoor activities restrictions, and reduction of social activities that seemed to be associated with an increased use of e-cig; also, the presentation at the emergency room amid pandemic, especially for young patients, required a rapid differential diagnosis between a severe viral infection and EVALI or VALI [7].

We included original (clinical) studies and excluded case reports, reviews, editorials, and experimental studies. Our strategy of research was based on the following combinations of “e-cigarette” (or “electronic cigarette”) and any of the following: “obesity”, “diabetes”, “metabolic”, “glucose”, “endocrine”, “hormone”, “glucocorticoid” and also a second cluster of research concerning various endocrine aspects such as “thyroid”, “pituitary”, “ovary”, “testes”, “fertility”, “infertility”, “sperm”, “adrenal”, “cortisol”, “osteoporosis”, “fracture”, and “bone”.

We checked the mentioned terms within the title and/or abstract, excluded the duplicates, and finally included only original (clinical) studies with clinical relevance (n = 22) (Figure 1).

## 3. Results

### 3.1. Diabetes Mellitus, Obesity, and Metabolic Syndrome

#### 3.1.1. Prediabetes

According to our methods, we identified three studies to address the issue of e-cigarettes and prediabetes, all of them having a group with dual e-cigarettes users and two included a specific subgroup with e-cigarette-only smokers [5,13,14]. Kim et al. [13] included 22,385 subjects (9490 men and 12,895 women) without diabetes, as following: 7.3% (N1 = 1628) of the studied population were dual smokers (e-cigarettes and conventional cigarettes), 31.1% (N2 = 6954) were cigarette-only smokers, 18.7% (N3 = 4181) were never smokers with exposure to second-hand smoking, and 43% (N4 = 9622) were never smokers without exposure to second-hand smoking. Prediabetes was diagnosed based on glycated hemoglobin (HbA1c) levels and had a prevalence of 24.1% in N1 group, respectively, of 33.1% in N2, 26.5% in N3, and 30.5% in N4. Prediabetes had higher odds ratio (OR) in dual users both *versus* never smokers without second-hand smoking (OR = 1.57, 95%CI: 1.29–1.92) and *versus* conventional cigarettes smokers (OR = 1.27, 95%CI: 1.07–1.52). There was no statistical significance with regard to the correlation between past cigarette-only use and prediabetes. The assessment based on gender revealed a higher risk of prediabetes in male dual smokers compared to non-smokers without second-hand smoking (OR = 1.70, 95%CI: 1.32–2.19), but no statistical significance for smoking behavior in women was registered [13].

A study conducted by Zhang et al. [14] included a population of 600,046 respondents, this study aimed to assess the prevalence of prediabetes and the association of smoking behavior based on self-reports. The prevalence of prediabetes was of 10.2% (95%CI: 9.8–10.7) in dual (e-cigarettes and conventional cigarettes) users [14] which is lower than Kim et al.’s findings [13]. One factor that might have influenced these different values of prevalence was the less precise method of prediabetes diagnosis used by Zhang et al. [14], meaning the fact that the prevalence of prediabetes was established upon asking the subjects if a doctor previously informed each of them as having the diagnosis of “prediabetes” or “borderline diabetes” [14]. In current e-cigarettes users the prevalence of prediabetes was of 9% (95%CI: 8.6–9.4), while in e-cigarette-only users the prevalence was of 5.9% (95% CI: 5.3–6.5). Unlike Kim et al.’s results [13], Zhang found no statistical significance for prediabetes and dual smoking (OR = 1.14, 95%CI: 0.97–1.34) [14].

Overall, these two studies pinpointed the fact that prediabetes might be associated, to a certain level to e-cigarettes use; one study showed that dual users have a higher risk; the other did not confirm it, and only one study included a specific e-cig-only subgroup. A 1.57-fold increase risk of having prediabetes has been found in individuals who were dual smokers *versus* those who were never smokers [13], thus raising the question of potential additive metabolic effects of e-cigarettes and cigarettes which are yet to be confirmed. Also, subjects who were non-smokers but have been dual smokers in the past still had an increased odd of prediabetes (a 1.27-fold increase) [13]. On the other hand, the third study conducted by Cai et al. [5] did not find a clear correlation between e-cig and prediabetes [5]. A pathogenic connection between prediabetes and e-cigarettes remains an open issue, while studies enrolling a higher number of participants are necessary to address the epidemiologic impact of prediabetes among e-cigarette users.

#### 3.1.2. Diabetes Mellitus 

Three studies addressed the issue of type 2 diabetes and e-cig. Of note, all the data we found only included this type of diabetes which is a traditional component of patients with various metabolic anomalies. Two studies from 2020 provided information about diabetes mellitus in e-cigarette users [15,16]. In a cross-sectional study on 7505 subjects, Kim et al. [15] showed that diabetes had a lower prevalence in dual smokers of both e-cigarettes and combustible cigarettes (3.3%) compared to cigarette-only smokers (5.9%); however, the prevalence was similar between dual smokers and never smokers, while dual smokers had a higher prevalence of diabetes family history than never-smokers [15].

A study on two data sets with a total of 7775 respondents (from 2015: N = 3627 and from 2018: N = 4148) was conducted by Leavens et al. [16]; the authors analyzed the prevalence of diabetes mellitus according to smoking behavior in adults experiencing homelessness. The rate of diabetes in e-cigarette-only users was of 13.4% (95%CI: 3.8–22.9), of 11.5% (95%CI: 8.4–14.5) in dual users, respectively. Diabetes mellitus was not associated with e-cigarette use. Moreover, the differences between e-cigarette-only use and non-smokers, conventional cigarettes-only smokers or dual smokers were not statistically significant; neither were those between dual smoking and conventional smoking or non-smoking. Diabetes had lower rates in combustible cigarette users compared to non-smokers: 10.9% (95%CI: 9.7–12) *versus* 13.6% (95%CI: 11.6–15.7). The fact that diabetes diagnosis was self-reported in a vulnerable population with limited access to healthcare was an important limitation of this study and it should be taken into consideration when analyzing these data [16]. As mentioned the study of Cai et al. [5] looked at the diabetes risk, as well; an e-cig user was more likely to present an increased index of insulin resistance (particularly, HOMA-IR) than a person who was never an e-cig user [5]. Current findings suggest a possible negative association between diabetes mellitus and dual smoking compared to combustible cigarette smoking, but the negative impact of e-cigarette use has not been specifically shown, probably the assessment of insulin resistance might provide additional clues of diabetes risk.

#### 3.1.3. Obesity 

We found six studies investigating a possible link between e-cigarettes and BMI and/or obesity [13,15,17,18,19,20], while three more studies provided data regarding the attainment of obesity prevention recommendations in e-cigarettes smokers [21], e-cigarettes use due to weight control purposes [22], and preferred e-cigarettes flavors in obese people [23]. The largest studied population was reported by Zhao et al. [17] on two cross-sectional data sets, including a total of 373,781 subjects (2015–2016: 189,306 individuals and 2018–2019: 184,475 persons), among whom 0.93% were past 30-day e-cigarettes users (2015–2016: 0.91% and 2018–2019: 0.96%). The weighted prevalence of 30-day e-cigarettes use was of 1.3% for 2015–2016 and of 1.6% for 2018–2019. E-cigarettes use had a higher prevalence in obese people in both data sets, with a weighted prevalence of e-cigarettes use of 1.8% (2015–2016), respectively, of 3.1% (2018–2019). Moreover, e-cigarettes use was associated with obesity in the general population (OR = 1.6, 95%CI: 1.3–2.1, p = 0.0007), in current conventional smokers (OR = 1.6, 95%CI: 1.2–2, p = 0.0005) and in people who had never smoked conventional cigarettes (OR = 2.4, 95%CI: 1.3–4.5, p = 0.0058) [17].

E-cigarettes use among obese individuals was reported by a large cross-sectional study published by Kim et al. [13]. The prevalence of dual smoking was higher in obese non-diabetic subjects compared to the general population (9.4% *versus* 7.3%) and in overweight subjects *versus* general population (7.4% *versus* 7.3%). Subjects with normal BMI had a lower prevalence of dual smoking than the general population (5.7% *versus* 7.3%). There was no available data regarding BMI and obesity in the group of e-cigarette-only smokers [13]. The relationship between dual smoking (e-cigarettes and conventional cigarettes) was also analyzed in another cross-sectional study (N = 7505 Korean subjects); 4.5% of the studied population were dual smokers, while 54.35% were cigarette-only smokers, and 40.33% were never-smokers, and 0.82% were e-cig-only smokers. Unfortunately, the group of e-cig-only use was not analyzed. Obesity had a higher prevalence in dual smokers (51%) *versus* cigarette-only (41.2%, p < 0.05) and never-smokers (39.2%, p < 0.05). A higher daily energy intake in dual smokers *versus* cigarette-only (p < 0.05) and never-smokers (p < 0.001) was also reported [15].

In a population-based birth cohort study conducted by Sompa et al. [18], 3055 subjects provided information regarding e-cigarettes/cigarette use. 3.9% (N = 120) of them smoked e-cigarettes, most of whom (77.5%) also added conventional cigarettes and/or snus. Only 22.5% of e-cigarettes smokers, accounting for 0.8% of the entire studied population (N = 27), smoked only e-cigarettes. BMI was assessed in 2265 subjects, waist circumference in 2251 subjects, and body fat % in 2229 subjects. An association between e-cigarettes and BMI, waist circumference, and body fat % was found only in female subgroups: women who were e-cigarette smokers had higher BMI (23.3 *versus* 22.1 kg/m^2^, p = 0.03), waist circumference (80 *versus* 73 cm, p = 0.003) and body fat % (30.3 *versus* 26%, p = 0.02) compared to non-smokers. In terms of overweight and obesity, BMI ≥ 25 kg/m^2^, waist circumference ≥80 cm for women and ≥93 cm for men and body fat ≥33% for women and ≥20% for men were significantly more frequent among e-cigarettes smokers. As limits of the study, there was no available data regarding potential differences between e-cig-only smoking and dual smoking of e-cig and conventional cigarettes [18].

Therefore, a potential association between obesity and e-cigarettes may be noted in terms of a higher prevalence of obesity and a lower prevalence of normal BMI in dual smokers compared to cigarette-only smokers and the general population [15], a higher prevalence of past 30-day e-cigarettes use in obese people [17], a higher prevalence of dual smoking in obese and overweight subjects, and a lower prevalence of dual smoking in subjects with normal BMI [13], a higher BMI, waist circumference, and % body fat in female e-cigarettes smokers [18], and a higher energy intake in dual smokers [15].

Other findings, however, were against the association between obesity and e-cigarettes use. Hoover et al. [19] did not find a statistically significant risk of obesity in e-cigarettes users despite a higher risk of food addiction (RR = 2.71, 99%CI: 1.75–4.21, p < 0.001) [19], while Alqahtani et al. [20] found that e-cigarettes use was associated with a lower BMI (B = −3.07, p = 0.021) [20].

Jacobs et al. [21] studied the attainment of obesity prevention guidelines in 12,578 teenagers, and reported that past 30-day e-cigarette-only smokers had higher attainment of physical activity and screen-time recommendations, while dual e-cigarettes and marijuana smokers did not meet sugar-sweetened beverages recommendations [21]. Mason et al. [22] investigated the weight control motives favoring e-cigarettes use in young adults (N = 99). The prevalence of obesity in the studied population was of 25%, while 26% of subjects were overweight. The study reported that e-cigarettes smoking for weight control were associated with sweet taste preference, especially at higher BMI [22]. Of note, another study by Mason et al. [23] revealed that overweight and obese subjects preferred menthol-flavored cigarettes [23].

#### 3.1.4. Metabolic Syndrome

Among the two studies addressing the metabolic syndrome in e-cig, Oh et al. [24] conducted a case-control study on 5462 individuals diagnosed with metabolic syndrome (45.9% males and 54.1% females) *versus* 12,194 controls (37.6% males and 62.4% females) and analyzed possible risk factors, including smoking, both conventional and electronic. Females smoking both conventional and e-cigarettes had higher odds of metabolic syndrome than non-smokers (OR = 4.02, 95%CI: 1.48–10.93, p = 0.006); while women smoking solely conventional cigarettes also had higher odds of metabolic syndrome than non-smokers (OR = 1.80, 95%CI: 1.02 *versus* 3.18, p = 0.042). Data for male subgroups were not statistically significant, except for the males with a history of >25 pack-years that had an increased risk of metabolic syndrome. The findings suggested a possible metabolic risk increase in female smokers of both conventional and e-cigarettes, but, as limits of the study we should mention a lack of a specific analysis of e-cigarettes use independently of the combustible cigarettes smoking. Therefore the effect of e-cigarettes on people who do not smoke combustible cigarettes could not be evaluated [24].

The prior mentioned study of Kim et al. [15] also included the results of metabolic syndrome and its components in 7505 subjects: the prevalence was higher in dual smokers compared to non-smokers (32.5% *versus* 23.2%, p < 0.001); the mean waist circumference was higher in dual smokers compared to cigarette-only and non-smokers (87.4 ± 0.6 cm *versus* 85.1 ± 0.2 cm, p < 0.001, respectively, 84.2 ± 0.2 cm, p < 0.001), with a higher prevalence of increased waist circumference in dual smokers compared to cigarette-only and non-smokers (39.3% *versus* 28.7%, p < 0.001, respectively, 25.5%, p < 0.001). Elevated triglycerides and low HDL-cholesterol had higher prevalence only in dual smokers compared to non-smokers (50.9% *versus* 30.8%, p < 0.001, and 36.0% *versus* 24.7%, p < 0.001). Fasting glucose levels were higher in dual smokers *versus* non-smokers (p < 0.001). High blood pressure, however, had a lower prevalence in dual smokers both compared to cigarette-only smokers (27.3% *versus* 40.2%, p < 0.001) and non-smokers (27.3% *versus* 34.0%, p < 0.05) [15].

The findings of these mentioned studies suggested a higher prevalence and odds of metabolic syndrome in dual smokers of e-cigarettes and combustible cigarettes, especially in females, while no controlled cohort specifically addressed a large population of only e-cig use, thus no clear conclusion can be established. Most components of metabolic syndrome (increased waist circumference, elevated triglycerides and reduced HDL-cholesterol) had a higher prevalence of dual smokers compared to non-smokers. When compared to cigarette smokers, dual smokers seemed to have a higher prevalence of increased waist circumference among the components of metabolic syndrome. Only high blood pressure had a lower prevalence in dual smokers compared to cigarette-only smokers and never-smokers [15,24] (Table 1).

### 3.2. Other (Non-Metabolic) Endocrine Elements in Relationship with E-Cig

#### 3.2.1. Potential Fertility Issues

Regarding the increasing use of e-cigarettes especially in teenagers and young adults in association with or as a substitute for combustible cigarettes, and considering different negative effects of e-cig exposure on the reproductive system, such as elevated oxidative stress and altered morphology, as found in animal models, it is essential to investigate whether humans are similarly affected by e-cigarettes [25,26,27]. According to our methods, we identified three studies that provided data regarding fertility; two of them addressed the issue of female fertility and e-cigarette use [28,29], and one investigated male fertility [30].

##### A. Female Fertility

Galanti et al. [28] investigated differences in fertility parameters in smoker and non-smoker women who underwent in vitro fertilization according to a prospective observational study. Out of 410 women suffering from idiopathic or tubal infertility that underwent intracytoplasmic sperm injection cycles, 51.5% were non-smokers or had ceased smoking for more than 1 year and 49.5% were active smokers. More than half of the smokers (51%) were cigarette smokers, and 29% of the women were e-cigarette smokers and 20% were using heat-not-burn products. Smokers had significantly lower AMH (Anti-Müllerian hormone) levels (p < 0.05), higher total dose of gonadotropin (p < 0.05), less oocytes retrieved per patient (p < 0.001), higher empty zona pellucid oocytes (p < 0.05), and lower fertilization rates (p = 0.03) when compared with non-smokers. Differences between smoker sub-types were not statistically significant, except for a higher number of germinal vesicles in e-cigarette smokers and heat-not-burn products than in cigarette smokers (0.48 ± 0.1 and 0.4 ± 0.1 *versus* 0.33 ± 0.2, p = 0.04). These data suggest that e-cigarette smoking might have a similar negative impact on fertility, especially on oocytes quality and quantity, as conventional cigarettes and heat-not-burn products [28].

Harlow et al. [29] conducted a pre-conception prospective cohort study on 4586 women that analyzed the effects of e-cigarette use on fertility ratio. E-cigarette use was reported in 17.1% of women (13.3% were former e-cigarette smokers and 3.8% were current e-cigarette smokers). Most e-cig smokers had combustible cigarette smoking history. Out of the former e-cigarette users, 25.2% were current cigarette smokers, 27.6% were former cigarette smokers, 9.9% were occasional cigarette smokers, and 37.3% never used cigarettes. Additionally, current e-cigarette users were current cigarette smokers in 19.5% of cases, former cigarette smokers in 45.3%, occasional cigarette smokers in 12.9%, and 23% never used cigarettes. Most women who did not use e-cigarettes, also never smoked combustible cigarettes (83%). However, e-cigarette use at baseline was associated with a non-significant reduction in fecundability before and after confounder adjustment, both in current users (fertility ratio adjusted for confounders of 0.85; 95%CI: 0.68–1.07) and former users (fertility ratio adjusted for confounders of 0.89; 95%CI: 0.78–1.00), with a fertility ratio adjusted for confounders in women who ever smoke e-cigarettes of 0.88 (95%CI: 0.78–0.99). The association between current e-cigarette use and lower fertility ratios in women who never smoked combustive cigarettes was not confirmed. Data regarding exclusive current e-cigarette use and dual current e-cigarette and cigarette use was also inconclusive. Overall, these studies did not clearly point out fertility impairment in females of reproductive age who use e-cigarettes; moreover, since fertility rate depends on a multifactorial panel, many other contributors should be taken into consideration [29].

##### B. Male Fertility

Holmboe et al. [30] analyzed a potential relationship between cigarette, e-cigarette, snuff, and marijuana smoking and male fertility in a cross-sectional study, including a total of 2008 young males. A total of 1043 cigarette smokers were compared with 946 non-smokers. They had statistically significant higher median levels of total testosterone (6.2% higher for daily smokers and 4.1% higher for occasional smokers, p < 0.01) and free testosterone (6.2% higher in both daily and occasional smokers, p < 0.01), and lower sperm concentrations (33 million/mL for daily smokers, 43 million/mL for occasional smokers and 44 million/mL for non-smokers, respectively, p < 0.01) and total sperm counts (103 million for daily smokers, 136 million for occasional smokers and 139 million for non-smokers, p < 0.01). A subgroup of 164 (25 daily and 139 occasionally) e-cigarette smokers were compared with 1057 non-smokers. A third of the e-cigarette smokers reported simultaneous cigarette use. There were no significant differences in terms of testosterone levels. Similarly to cigarette smokers, young males who were e-cigarette smokers had lower sperm concentrations (33 million/mL for daily smokers, 39 million/mL for occasional smokers, and 45 million/mL for non-smokers, p < 0.01) and total sperm counts (91 million for daily smokers, 128 million for occasional smokers and 147 million for non-smokers, p < 0.01) than non-smokers. This difference remained irrespective of concomitant cigarette use. These findings suggested a negative effect on male fertility, namely lower sperm counts and concentrations, in e-cigarette users, but the current number of similar studies remains limited [30].

Overall, data regarding fertility and e-cigarette use was scarce. In women, a negative impact on fertility was found in terms of reduced oocytes quality and quantity in one study [28], and lower fertility ratios in another study [29]. However, the first mentioned study did not find differences between e-cigarette smokers and combustible cigarette users, while the second mentioned study did not find conclusive data regarding e-cigarettes effects in women who never smoked combustible cigarettes. Therefore, it is difficult to assess the risk of e-cigarette use on female fertility due to a tight association with combustible cigarette use. Regarding male fertility, similarly to combustible cigarette use, e-cigarette smoking was associated with lower sperm counts and concentrations according to a single clinical study [30]. Further research will pinpoint the fertility impact amid e-cig. (Table 2).

#### 3.2.2. EVALI and the Use of Systemic Steroids

Other potential metabolic/endocrine interferences are originating from the systemic steroids therapy as part of the EVALI management. COVID-19 pandemic burst somehow diverted the diagnosis of EVALI and severe acute forms of the infectious disease shared some features at first admission, especially with respect to pulmonary complications [31,32,33]. However, data on EVALI were reported, too, during pandemic years and we identified nine studies concerning the use of steroids as part of EVALI management within our time frame of research. EVALI represents a severe inflammatory syndrome that was described for the first time in 2019 at teenagers who were admitted for different manifestations of respiratory insufficiency in relationship with e-cigarette use [34].

A retrospective study based on electronic records of teenagers with EVALI who were hospitalized before (N = 19) and during COVID-19 pandemic (N = 22) showed on 39/41 patients that needed glucocorticoids therapy a clinical improvement in 79% (31/39) of cases within the first 24 h; subjects admitted during pandemic associated a shorter duration of hospitalization when compare to pre-pandemic era (a median of 5 days *versus* 7 days, p < 0.01). Of note, EVALI might be missed during severe presentations of coronavirus infection, while anamnesis should mandatory include the history of e-cigarette/ENDS use nowadays [35]. Moreover, the selective use of glucocorticoids in COVID-19 was lifesaving in selected cases such as severe acute respiratory distress syndrome or COVID-19 pneumonia, but their use is necessary for EVALI, too [36,37].

Also, a 1-year small study (N = 8) on adolescents that were prior diagnosed with EVALI (and all received corticoid therapy) showed, however, that, after experiencing such a severe respiratory episode, all improved the radiographic manifestations and spirometry parameters at short-term evaluation (an average follow-up of 46.5 days) suggesting, not only a certain degree of reversibility, but also the important role of offering glucocorticoids. That is why the potential panel of side effects should be taken into consideration, as well [38]. For what we know so far, EVALI treatment mandatory includes glucocorticoid therapy for acute lung injury and discontinuation of vaping/e-cigarette [39]. Similarly, Corcoran et al. [40] reported a small cohort of 7 teenagers (with a median age of 17) who required systemic steroids (42%) amid admission for EVALI (which associated bilateral opacities at radiological lung assessment, respiratory and non-respiratory complications such as gastrointestinal in 57% of them) [40,41].

Particularly, adolescents seem prone to develop EVALI; a short course of systemic steroids is associated with a fewer side effects than prolonged glucocorticoids exposure, but the potential negative effects should be considered, including the impact on peak bone mass [42]. However, there are studies [41,43] such as the cohort published by Carroll et al. [43] showing that the need of long term inhalator administration after an acute episode is sometimes required thus it increases the risk of steroids - associated side effects (N = 15 patients with a mean age of 17.1 years who suffered an episode of EVALI; in this study, after the systemic use of steroids during an acute episode with a clinical improvement, 5/11 of them continued with inhaled glucocorticoids) [43].

Apart from teenagers, EVALI also affects adults, of ages up to 62 years. The rates of glucocorticoid requirement are seemingly lower, of 50%, with a good clinical outcome [44]. Most often patients require both oral and intravenous glucocorticoids. In a retrospective study on 13 teenagers suffering from EVALI, 12/13 subjects required steroids. The vast majority (83%) received both oral and intravenous glucocorticoid treatment, while only one patient received oral glucocorticoids alone and another was treated with intravenous glucocorticoids [45]. Prednisone equivalent daily doses, as reported by one of the largest studies on EVALI, varied between 40 and 71 mg, with a median of 46 mg [46] (Table 3).

## 4. Discussion

A few years gap has been registered between the hopeful introduction of e-cigarettes/other ENDS and the raise of a potential side effects panel, particularly in relationship with a high systemic cytokines profile and cardiopulmonary injury. A heterogeneous spectrum of aspects may be analyzed in relationship with e-cigarettes, while the larger picture of e-cig is situated at the crossroad between daily habits, lifestyle intervention, social burden, and medical issues [47,48,49,50,51,52,53,54,55,56,57,58].

### 4.1. E-Cig: Is It Time for Taking into Consideration a New Type of Endocrine Disruptor?

The multitude of components found in e-cigarettes, some acting as potential endocrine disruptors [59], as well as the possible interactions between their intrinsic constituents, makes it difficult to predict the effect of e-cigarettes on metabolic profile as well as fertility [60,61,62,63,64]. Whether exposure to different types of e-cig might negatively impact the endocrine system and glucose metabolism is yet to be proven, but currently this remains an open issue that is mandatory to be further explored noting the massive spreading of e-cig use amid modern era, especially in teenagers and young adults.

### 4.2. E-Cig and Data with Regard to Prediabetes and Diabetes

E-cigarettes are sometimes used as an aid against weight gain in people seeking to cease smoking, due to the continuing effects of nicotine, including appetite suppression [9], increased sympathetic activation, and increased thermogenesis [10]. However, nicotine may also promote abdominal obesity and increased insulin resistance [10]. Moreover, the rates of success with concern to smoking cessation might not be so high [11,12]. In order to properly assess the risk-benefit balance, understanding the metabolic profile of the e-cigarette users seems like an important aspect. As mentioned, we need to look at the most important metabolic components with respect to e-cig use, from type 2 diabetes mellitus to milder glucose profile anomalies (such as prediabetes) to obesity, insulin resistance, and entire panel of metabolic syndrome [65,66,67,68,69,70,71,72,73].

The studies regarding prediabetes and e-cigarettes showed a possible link, yet, not all agree [5,13,14]. The higher rate of prediabetes might be caused by multiple factors, including nicotine, which was shown to increase insulin resistance [74], the sugar content, and flavors of the liquid [75]. Apart from the effect of nicotine, the liquid itself might impair glucose metabolism, as suggested by a study on rats, nicotine-free e-cigarette liquid induced hyperglycemia [76]. Even though both prediabetes and metabolic syndrome were studied in relationship with e-cigarette use, the prevalence of diabetes was lower in one of the studies [15], and another did not find any statistically significant association [16] thus a clear conclusion cannot be established. One possible explanation behind these findings might be the young age of e-cigarette users that was associated with dual smoking in Kim et al.’s study [15]. In Leavens et al.’s study [16], conducted on adults experiencing homelessness, the e-cigarette users were younger than non-users. Moreover, e-cigarette users had a higher frequency of lifetime homelessness. The prevalence of diabetes might be underestimated in this study due to self-reported diagnosis [16]. Of note, the type of prediabetes assessment might bring a potential bias regarding its prevalence.

Considering that data regarding the association between e-cigarettes and diabetes is limited and that most users are younger than conventional cigarette smokers, the possible risks posed by e-cigarette smoking in diabetic patients should not be underestimated. One example of such a risk is provided by data from mice models suggesting that e-cigarette smoke impairs wound healing and angiogenesis, especially when having the background of diabetes [77].

### 4.3. E-Cig Users and Obesity

Most studies found either an association between e-cigarettes and obesity or a higher prevalence of obesity among e-cigarette users (4 out of 6 studies) [13,15,17,18]. The outcomes of one study included a higher risk of food addiction in e-cigarette users, but not a statistically significant association with obesity [19]. A negative relationship between BMI and e-cigarette use was also reported [20]. The profile of the obese patient might favor e-cigarette use due to weight control reasons, to prevent binge eating considering nicotine’s appetite suppression effects [65], and to satisfy cravings using the large flavor palette available in these products [22,23,66]. Oftentimes, e-cigarette smoking is associated with eating disorders, including not only binge eating, and bulimia nervosa, but also anorexia [67], especially in adolescent girls [68,69]. Of note, Sompa et al. [18] found an association between higher BMI, waist circumference and % body fat with dual smoking only in females, but not in males [18]. Physical inactivity has also been linked to e-cigarette use in teenagers [70]. While the aforementioned facts are arguments for a possible link to obesity, the association with anorexia [67], unhealthy weight control behaviors such as using laxatives and diet pills, and ingesting fewer calories [71], as well as compulsive exercise [72] favor a lower BMI. In vitro studies, however, do not support body weight changes as an effect of e-cigarettes [73]. The relationship between e-cigarettes and obesity seems multivalent and requires further investigations.

### 4.4. Metabolic Syndrome among E-Cig Users

Conventional cigarette smoking has previously been linked to metabolic syndrome [78] while quitting has been found to decrease its odds [79]. One of the main pathogenic pathways involves nicotine. Nicotine can promote a hyperglycemic state by decreasing tissues sensitivity to insulin and it can also increase free fatty acids by promoting lipolysis and by preventing lipogenesis [80]. Regarding the impact of e-cigarettes on metabolism, current findings showed an increased prevalence of metabolic syndrome in dual smokers [15] and increased odds of metabolic syndrome in females [24]. When analyzing the independent components of metabolic syndrome, Kim et al.’s [15] study reported that most components were independently linked to dual smoking. Interestingly though, high blood pressure had a lower prevalence in dual smokers [15]. In animal models, e-cigarettes have been linked to arterial stiffness, and vascular endothelial changes [81], while in humans acute inhalation of e-cigarette smoke might cause high blood pressure [82]. However, in Asian populations, conventional smoking has been associated with low systolic and diastolic blood pressure [83]. Another potential contributor to a lower prevalence of high blood pressure might be the younger age of dual smokers. The impact of e-cigarettes alone has not been studied. The metabolic implications of e-cigarette smoking, particularly on glucose metabolism, remain an open subject. The outcome of quitting conventional cigarettes, while still using e-cigarettes is a subject of great interest and requires further studies in order to establish whether e-cigarettes alone may pose quantifiable risks to metabolism.

### 4.5. Fertility Profile Regarding E-Cig

Current data on female fertility suggest a negative impact with lower fertility rates and reduced oocyte quality and quantity. Still, e-cigarettes’ influence cannot be unlinked from the effect of the exposure to combustible cigarettes. Even though Galanti et al. [28] found a negative impact on oocyte quality and quantity in smokers, and no statistically significant difference between smoking subtypes, e-cigarettes were not independently compared to non-smoking [28]. The findings of Harlow et al. [29] linked e-cigarette use to a discrete reduction in fertility rates, but the effect did not remain in women who never smoked combustible cigarettes [29]. Regarding male fertility, the findings of Holmboe et al. [30] are consistent with findings in animal models and indicate lower sperm counts and concentrations in males using e-cigarettes [30]. The topic of fertility in the context of e-cig use is still an open issue and it requires further studies on humans to investigate whether the biological processes in adults exposed to electronic cigarettes are consistent with findings from animal models. Another connected topic in this specific matter (we could not identify any clinical study according to our methods) involves the use of oral contraceptives in e-cigarettes smokers and potential increase of thromboembolism risk as reported, for instance, by one case published in 2023 [64].

Apart from nicotine’s known effects on reproduction such as the disruption of the hypothalamic-pituitary gonadal axis, the increased oxidative stress in the testis and altered sperm quality and quantity, and impairment of fertilization, as determined by the studies on conventional cigarettes, other constituents might alter fertility. Heavy metals, flavors, including bubble gum, cinnamon, and juice flavor, as well as vapors containing formaldehyde and volatile organic compounds, have been associated with altered fertility in male and female animal models [60,61]. Findings from rats exposed to nicotine-free products are further clues that nicotine is not the sole factor to alter fertility. E-cigarette refill liquid was found to decrease sperm quality and density, increase oxidative stress, create an inflammatory state in the epididymis, and disrupt steroidogenesis in male rats, even in nicotine-free products [62,63]. Concerning female fertility, animal models have found impairment of ovarian function and implantation, but not in oocyte quality [60], while data on humans are still a matter of unknown in some areas [64].

### 4.6. Thyroid Assessment in Subjects Using E-Cig

Other endocrine areas we searched did not pinpoint consistent original studies. A possible relationship with thyroid pathology was suggested in Aranyosi et al.’s [84] study on orbital fibroblast cultures exposed to different cigarette smoke types. Graves’s eye disease-associated orbital fibroblasts exposed to e-cigarette smoke had the highest change in cell index, yet in vivo effects of e-cigarette smoke remain unclear and it is suggested that there might be differences from the effects of conventional cigarette smoke [84]. Of note, we mention the importance of thyroid nodules’ evaluation in general population as the most frequent endocrine condition [85]; yet, a specific clinical perspective in e-cig users is not established in relationship with this specific frame.

### 4.7. Bone Status, including Peak Bone Mass, in Teenagers and Young Adults Using E-Cig

Considering that most e-cigarette users are teenagers and young adults who are about to reach their peak bone mass, the effect of e-cig on bone metabolism is of great interest, but current data regarding this subject are lacking [86]. Conventional cigarette smoking, however, has previously been studied and linked to disturbed bone metabolism and lower bone mineral density and higher fracture risk including in younger individuals [87]. Moreover, smoking at younger ages impairs peak bone mass acquisition during puberty [88]. The question whether e-cigarettes lead to similar negative outcomes as conventional smoking emerges. Even though clinical studies are deficient, animal and in vitro studies provided some insights into this subject. Two animal models reported different effects of e-cigarette smoke on bone. Reumann et al.’s [89] study on mice bone structure revealed a bone-preserving effect of e-cigarette smoke compared to conventional cigarette smoke [89]. A study on zebrafish embryos found that e-cigarette liquid and flavoring perturb bone development [90]. In vitro studies on osteoblasts have linked e-cigarette vapors to increased oxidative stress [91], reduced viability, and impaired function [92], suggesting possible negative outcomes on fracture healing and bone acquisition. These hypotheses need to be tested in humans.

As mentioned, glucocorticoids treatment for EVALI as a possible risk factor for impaired bone acquisition is not to be neglected, as similarly seen in other medical circumstances [93]. Considering that most users are young, the impact of e-cigarette smoking on bone acquisition and peak bone mass, as well as the impact of glucocorticoid treatment for EVALI represent important future research directions. Moreover, our research did not identify any papers related to pituitary, adrenals and parathyroid glands. Due to the young age of most e-cig users, it is advisable to include in the interview (anamnesis) questions related to the use of such products, especially considering the potential disturbances of fertility and metabolism. Also, murine experiments showed that nicotine from e-cigarettes impairs male’ muscle contractility and running speed while propylene glycol and vegetable glycerin (the components of the vehicle that help the nicotine delivery) damages the recovery of muscle injury. Additionally, nicotine from inhaled aerosol alters catecholamine levels [94].

### 4.8. Cannabinoid Receptors

The activation of autonomic nervous system has been associated with nicotine use and its alternatives. It has suggested that cannabidiol/tetrahydrocannabidol binds cannabinoid receptors, but also central dopamine and serotonin receptors inducing psychoactive effects; moreover, these substances may interact with usual medication such as anticoagulants, etc. and further studies are necessary to pinpoint these interferences [95,96]. Possible influences of e-cigarettes on catecholamine levels might be mediated by nicotine, which stimulates the release of catecholamines [97]. Evidence regarding their blood levels, however, did not show higher values. In a study on 36 healthy dual smokers exposed to different smoking conditions (cigarette smoking, e-cigarette smoking and non-smoking), heart rate, blood pressure and various biomarkers of oxidative stress were analyzed and the results showed that the heart rate was higher in cigarette smoking compared to e-cigarettes, and also in e-cigarette users *versus* non-smokers. Blood pressure had similar rates in cigarette smoking and e-cigarette smoking-associated conditions, both conditions leading to higher blood pressure values than non-smoking conditions. Despite different circadian hemodynamic effects, 24-h urinary catecholamine levels did not have statistically significant differences across different smoking conditions [98].

Whether the crossroads with cannabinoid system and catecholamines status are essential players in the metabolic and endocrine side effects of e-cig is still a matter of further studies. Other aspects are yet to be understood. For instance, we mention an interesting aspect that is also less clear nowadays: an acute hypersensitivity reaction was suspected to inhale antigen on a 16-year-old female who was confirmed with idiopathic acute eosinophilic pneumonia after she used e-cigarettes for 1 year (tetrahydrocannabinol cartridge) thus suggesting potential (antigen-induced) autoimmune effects [99].

### 4.9. Further Expansion of E-Cig-Related Endocrine Researches

As limits of the current work we mention the narrative review type, but the heterogeneous panel of parameters that have been addressed among the mentioned studies could not be limited to a selective number of criteria according to a systematic review. The time frame was specifically chosen to cover the pandemic years upon specific guidance in e-cigarettes and other ENDS has been released with regard to pulmonary complications (in 2019) [100,101].

Overall, this research seems like an early highlight of growing medical evidence that is expected to be released due to a larger use of e-cigarettes. This review is, to our aware, one of the most complexes of its type concerning potential endocrine and metabolism disorders that were linked to the e-cig use as part of current lifestyle choices.

Further controlled, longitudinal studies are necessary to pinpoint various endocrine issues, not only in adolescents and young adults, but, also, in older patients. Moreover, new data are required to assess the influence of prior/concomitant conventional smoking *versus* e-cig.

## 5. Conclusions

Overall, the most important metabolic highlights within the mentioned studies showed that among dual smokers there is a 1.57-fold prediabetes increase risk compare to those never smoking, with an increased prevalence of 10.2% (dual) *versus* 5.9% (e-cig-only). Current non-smokers who have been dual smokers still have an increased odd of prediabetes (a 1.27-fold risk increase). A lower prevalence of type 2 diabetes among dual users (3.3%) *versus* cigarette smoking (5.9%) was identified. One study found that e-cig use was associated with obesity in general population; another showed that obesity had a higher prevalence in dual smokers (51%) *versus* cig-only (41.2%, p < 0.05), while another found that female (not male) e-cig smokers had higher BMI. Data on metabolic syndrome are provided for dual smokers: one case-control study found that female (not male) dual smokers had higher odds of this syndrome than non-smokers. The need for awareness with respect to potential e-cigarette-associated medical issues should be part of modern medicine, including daily anamnesis. Whether metabolic/endocrine frame is part of this general picture is yet to be determined. Surveillance protocols should help the clinicians to easily access the medical background of one subject, including this specific matter of electronic cigarettes with/without conventional cigarettes smoking and other habits/lifestyle elements, especially when taking into consideration metabolism anomalies.

## Figures and Tables

**Figure 1 metabolites-13-01192-f001:**
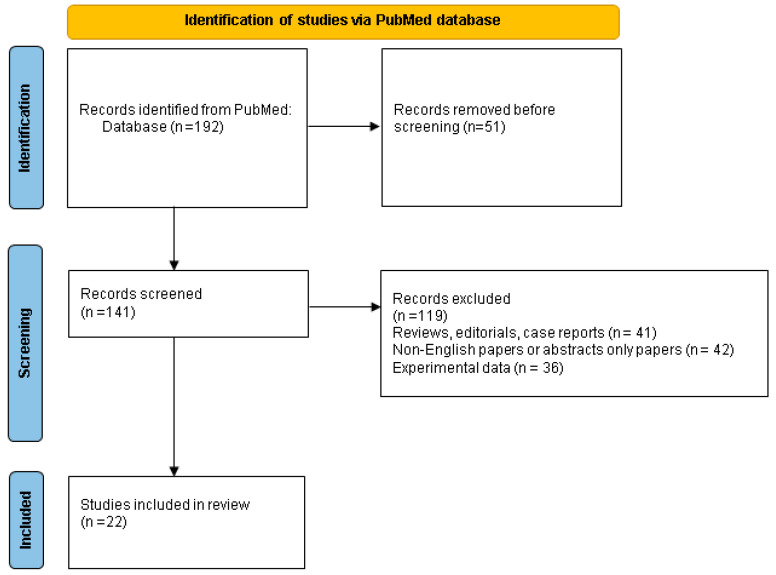
Flowchart of PubMed research according to the mentioned key words and study designs.

**Table 1 metabolites-13-01192-t001:** Studies on metabolic issues and e-cigarettes according to our methods regarding: prediabetes [13,14], diabetes mellitus [15,16]; obesity [13,15,17,18,19,20]; metabolic syndrome [15,24].

First Author Year of Publication Reference Number	Study Design and Population	Key Findings
**Prediabetes**		
Cai 2023 [5]	Cross-sectional study (NHANES) * N = 5101 adults from U.S. N1 = 6.3% current e-cig users N2 = 17.1% former e-cig users	E-cig use was not correlated with prediabetes (p > 0.05) – fully adjusted model No correlation between dual use and prediabetes (p > 0.05)
Kim 2022 [13]	Cross-sectional (nationwide population-based) study ** N = 22,385 subjects (9490 men and 12,895 women) without diabetes (a subgroup of 6735 patients had prediabetes) Studied groups: N1 = 1628: dual smokers (e-cigarettes and conventional cigarettes): 7.3% N2 = 6954: cigarette-only smokers: 31.1% N3 = 4181: never smokers with exposure to second-hand smoking: 18.7% N4 = 9622: never smokers without exposure to second-hand smoking: 43%	Prevalence of prediabetes: N1: 24.1%; N2: 33.1%; N3: 26.5%; N4: 30.5% OR of prediabetes: N1 *versus* N3: OR = 1.57 (95%CI: 1.29–1.92) N1 *versus* N2: OR = 1.27 (95%CI: 1.07–1.52) Current dual users with dual past use: OR = 1.54 (95%CI: 1.09–2.18) Current single conventional cigarettes users with dual past use: OR = 1.67 (95%CI: 1.31–2.13) Current non-smokers with dual past use: OR = 1.54 (95%CI: 1.04–2.31) No statistical significance for current non-smokers with single past conventional cigarettes use. Higher odds of prediabetes in men (but not for females) who were dual users *versus* never-smokers without exposure to second-hand smoking: OR = 1.70 (95%CI: 1.32–2.19)
Zhang 2022 [14]	Cross-sectional study 600,046 subjects	Prevalence of prediabetes in: Current e-cigarettes users: 9% (95%CI: 8.6–9.4) E-cigarette-only users: 5.9% (95%CI: 5.3–6.5) Dual (e-cigarettes and conventional cigarettes) users: 10.2% (95%CI: 9.8–10.7) Higher odds of prediabetes compared to never e-cigarette users in: Current e-cigarette users: 1.22 (95%CI: 1.1–1.37) Former e-cigarette users: 1.12 (95%CI: 1.05–1.19) E-cigarette-only users: 1.54 (95%CI: 1.17–2.04) No statistical significance for dual users.
**Diabetes mellitus**		
Cai 2023 [5]	*	E-cig use was not correlated with diabetes (p > 0.05)—fully adjusted model No correlation between dual use and diabetes (p > 0.05) No correlation between dual use and insulin resistance (p > 0.05) N1 were 63% (95%CI: 1–2.91) more likely to have higher HOMA-IR than never e-cig users N2 were 64% (95%CI: 1.04–2.59) more likely to have higher HOMA-IR than never e-cig users
Kim 2020 [15]	Cross-sectional study N = 7505 subjects: *** dual smokers: 4.5% (N1 = 337) cigarette-only: 54.35% (N2 = 4079) never smokers: 40.33% (N3 = 3027) e-cigarettes only: 0.82% (N4 = 62)	Lower prevalence of diabetes in dual smokers (3.3%) *versus* cigarette-only (5.9%) (p < 0.05) Similar prevalence of diabetes in dual smokers *versus* never smokers (p > 0.05) Higher prevalence of diabetes family history in dual smokers (22.2%) *versus* never smokers (15.4%) (p < 0.05)
Leavens 2020 [16]	Cross-sectional study N total = 7775 respondents **2 data sets:** **2015**: N = 3627 adults experiencing homelessness e-cigarette users: 11.6% (N1 = 421) combustible cigarette users: 67.7% (N2 = 2456) dual users: 10.4% of all participants and 89.5% of e-cigarette users (N = 377) **2018:** N’ = 4148 adults experiencing homelessness e-cigarette users: 14.6% (N1′ = 607) combustible cigarette users: 72.8% (N2′ = 3021) dual users: 13% of all participants and 88.8% of e-cigarette users (N3′ = 539)	Diabetes was not associated with e-cigarette use, neither in e-cigarette only, nor in dual smokers. Diabetes had lower rates in combustible cigarette users compared to non-smokers: 10.9 (95%CI: 9.7–12) *versus* 13.6 (95%CI: 11.6–15.7)
**Obesity**		
Sompa 2022 [18]	Population-based birth cohort N = 3055 subjects e-cigarettes smokers: 3.9% (N1 = 120) 77.5% (N1′ = 93) also used conventional cigarettes and/or snus 22.5% (N1′’ = 27) smoked only e-cigarettes BMI assessed in 74.14% (N2 = 2265) subjects Waist circumference assessed in 73.68% (N3 = 2251) subjects Body fat % assessed in 72.96% (N4 = 2229) subjects	Current e-cigarettes female smokers had: Higher BMI: 23.3 (20.6, 28.7) *versus* 22.1 (20.2, 24.0) kg/m^2^, p = 0.03 Higher waist circumference: 80.0 (71.5, 89.5) *versus* 73.0 (69, 79) cm, p = 0.003 Higher body fat %: 30.3 (23.2, 36.1) *versus* 26.0 (22.4, 30.3) %, p = 0.02 Overall, e-cigarettes smoking was associated with: Overweight/obesity: OR = 1.8 (95%CI: 1.0–3.2) Waist circumference ≥80 cm for women and ≥93 cm for men: OR = 1.9 (95%CI: 1.0–3.4) Body fat ≥33% for women and ≥20% for men: OR = 2.6 (95%CI: 1.4–4.6)
Kim 2022 [13]	**	Prevalence of dual smoking is higher in: Obese subjects *versus* general population (9.4% *versus* 7.3%) Overweighed subjects *versus* general population (7.4% *versus* 7.3%) Lower in normal weighted subjects *versus* general population (5.7% *versus* 7.3%)
Hoover 2022 [19]	Community sample study N = 357 adults e-cigarettes use: 7.8%	Risk of food addiction was higher in e-cigarettes smokers: adjusted RR = 2.71, (99%CI: 1.75–4.21) p < 0.001 Risk of obesity was not statistically significant in e-cigarettes smokers: adjusted RR = 0.64 (99%CI: 0.1–4.11) p = 0.539
Zhao 2020 [17]	Two cross-sectional **data sets**: **2015–2016:** N = 189,306 people 0.91% (N1 = 1725) e-cigarettes users (weighted prevalence of e-cigarettes use = 1.3%) **2018–2019:** 184,475 people 0.96% (N2 = 1777) e-cigarettes users (weighted prevalence of e-cigarettes use = 1.6%) **Total:** 373,781 people 0.93% (N’ = 3502) e-cigarettes users	Weighted prevalence of e-cigarettes smoking is higher in obese people: 2015–2016 = 1.8% (1.4 to 2.2) 2018–2019 = 3.1% (2.2 to 4.1) E-cigarettes use associated with obesity in: the general population: OR = 1.6 (95%CI: 1.3–2.1) p = 0.0007 current conventional smokers: OR = 1.6 (95%CI: 1.2–2.0) p = 0.0005 never smokers of conventional cigarettes: OR = 2.4 (95%CI: 1.3–4.5) p = 0.0058
Alqahtani 2020 [20]	Secondary data analysis of 207,117 electronic medical records → a sample of 965 patients (Current e-cigarettes users: 5%)	E-cigarettes use associated with a lower BMI (B = −3.07, p = 0.021)
Kim 2020 [15]	***	Higher daily energy intake in dual smokers *versus* cigarette-only (p < 0.05) and never smokers (p < 0.001) Lower prevalence of normal BMI in dual smokers (26.1%) *versus* cigarette-only (35.1%, p < 0.05) and never smokers (35.5%, p < 0.05). Lower prevalence of pre-obesity in dual smokers (22.9%) *versus* cigarette-only (23.8%, p < 0.05) and never smokers (25.3%, p < 0.05). Higher prevalence of obesity in dual smokers (51.0%) *versus* cigarette-only (41.2%, p < 0.05) and never smokers (39.2%, p < 0.05).
**Metabolic syndrome**		
Kim 2020 [15]	***	Dual users *versus* non-smokers: Higher prevalence of metabolic syndrome: 32.5% versus 23.2%, p < 0.001 Higher mean waist circumference: 87.4 ± 0.6 cm *versus* 84.2 ± 0.2 cm, p < 0.001 Higher prevalence of increased waist circumference: 39.3% *versus* 25.5%, p < 0.001 Lower prevalence of high blood pressure: 27.3% *versus* 34.0%, p < 0.05 Higher mean fasting glucose: 100.9 ± 1.5 mg/dL *versus* 97.5 ± 0.5 mg/dL, p < 0.001 Higher prevalence of high triglycerides: 50.9% *versus* 30.8%, p < 0.001 Higher prevalence of low HDL-cholesterol: 36.0% *versus* 24.7%, p < 0.001 Dual users *vs.* cigarette-only smokers: No statistically significant difference in prevalence of metabolic syndrome Higher mean waist circumference: 87.4 ± 0.6 cm *versus* 85.1 ± 0.2 cm, p < 0.001 Higher prevalence of increased waist circumference: 39.3% *versus* 28.7%, p < 0.001 Lower prevalence of high blood pressure: 27.3% *versus* 40.2%, p < 0.001 Dual users *versus* never smokers had higher adjusted prevalence OR for: Metabolic syndrome: 2.79 (95%CI: 1.72–4.53, p < 0.001) Elevated waist circumference: 2.26 (95%CI: 1.31–3.91, p = 0.003) Elevated triglycerides: 2.81(95%CI: 1.90–4.14, p < 0.001) Reduced HDL-cholesterol: 2.48 (95%CI: 1.66–3.71, p < 0.001) Dual users *versus* cigarette-only smokers had higher adjusted prevalence OR for: Metabolic syndrome: 1.57 (95% CI = 1.03–2.40, p = 0.038) Elevated waist circumference: 1.96 (95% CI = 1.19–3.23, p = 0.008) Reduced HDL-cholesterol: 1.90 (95% CI = 1.31–2.76, p = 0.001)
Oh 2020 [24]	Case-control study N = 5462 cases of metabolic syndrome (2507 males and 2955 females) N’ = 12194 controls (4585 males and 7609 females)	Increased odds of metabolic syndrome in: Females who were current smokers of conventional cigarettes OR = 4.02 (95%CI: 1.48–10.93, p = 0.006) Females who were current smokers of conventional and e-cigarettes OR = 1.80 (95%CI: 1.02–3.18, p = 0.042)

Abbreviations: BMI = body mass index; CI = confidence interval; HDL = high density lipoprotein; N = number of patients; OR = odd ratio; RR = relative risk; *, **,*** the same studied population.

**Table 2 metabolites-13-01192-t002:** Studies on fertility profile in e-cigarettes users according to our methods [28,29,30].

First Author Year of Publication Reference Number	Study Design	Key Findings
Galanti 2023 [28]	Prospective, observational study N = 410 infertile women N1 = 203 (51.5%) non-smokers N2 = 207 (49.5%) active smokers N3 = 103 (51%) cigarette smokers N4 = 60 (29%) e-cigarette smokers N5 = 40 (20%) heat-not-burn products	Lower AMH in smokers (1.3 ± 2.3 ng/mL *versus* 2.1 ± 4.5 ng/mL, p < 0.05) Higher total dose of gonadotropin in smokers (1850 ± 860 UI *versus* 1730 ± 780 UI, p < 0.05) Less oocytes retrieved/patient in smokers (5.21 ± 0.9 *versus* 6.55 ± 3.5, p < 0.001) Higher empty zona pellucid oocytes in smokers (0.51 ± 0.1 *versus* 0.2 ± 0.1, p < 0.05) Lower fertilization rate in smokers (68.12 ± 2.21 *versus* 72.16 ± 3.05, p = 0.03) More germinal vesicles in e-cigarette smokers and heat-not-burn products compared with cigarette smokers (0.48 ± 0.1 and 0.4 ± 0.1 *versus* 0.33 ± 0.2, p = 0.04)
Harlow 2021 [29]	Prospective cohort study N = 4586 women trying to conceive At baseline: 83% never used e-cigarettes (83.1% of them never used combustible cigarettes) 13.3% former e-cigarettes users (25.2% current cigarettes smokers, 27.6% former cigarettes smokers, 9.9% occasional cigarette smokers, 37.3% never used cigarettes) 3.8% current e-cigarettes users (19.5% current cigarettes smokers, 45.3% former cigarettes smokers, 12.9% occasional cigarette smokers, 23% never used cigarettes)	Small reduction in fecundability before and after confounder adjustment, both in current and former users: Ever e-cigarette smokers at baseline: fertility ratio adjusted for confounders = 0.88 (95%CI: 0.78–0.99) No statistical significance for: Current e-cigarette smokers at baseline: fertility ratio adjusted for confounders = 0.85 (95%CI: 0.68–1.07) Former e-cigarette smokers at baseline: fertility ratio adjusted for confounders = 0.89 (95%CI: 0.78–1.00) Women who never smoked combustible cigarettes: fertility ratio adjusted for confounders = 0.87 (95%CI: 0.57–1.32) Current e-cigarette use: fertility ratio adjusted for confounders = 0.91 (95%CI: 0.7–1.18) Dual current e-cigarette and cigarette use: fertility ratio adjusted for confounders = 0.83 (95%CI: 0.54–1.29)
Holmboe 2020 [30]	Cross-sectional study N = 1043 (467 daily and 576 occasionally) cigarette smokers, compared with 946 non-smokers N’ = 164 (25 daily and 139 occasionally) e-cigarette smokers, compared with 1057 non-smokers	Higher total (6.2% higher for daily smokers and 4.1% higher for occasional smokers, p < 0.01) and free testosterone (6.2% higher in both daily and occasional smokers, p < 0.01) rates in cigarette smokers. Lower sperm concentrations (33 million/mL for daily smokers, 43 million/mL for occasional smokers and 44 million/mL for non-smokers, respectively, p < 0.01) and total sperm count (103 million for daily smokers, 136 million for occasional smokers and 139 million for non-smokers, p < 0.01) in cigarette smokers. Lower sperm concentrations (33 million/mL for daily smokers, 39 million/mL for occasional smokers and 45 million/mL for non-smokers, p < 0.01) and counts (91 million for daily smokers, 128 million for occasional smokers and 147 million for non-smokers, p < 0.01) in e-cigarette smokers.

Abbreviations: AMH = Anti-Müllerian Hormone; CI = confidence interval; N = number of patients; OR = odd ratio.

**Table 3 metabolites-13-01192-t003:** Studies with reports of systemic steroids use for EVALI according to our methods [35,38,39,40,41,43,44,45,46].

First Authors Year of Publication Reference Number	Study Design	End Points
Abdallah 2023 [35]	Retrospective study N = 41 teenagers with EVALI N1 = 19 subjects admitted before COVID-19 pandemic N2 = 22 subjects admitted during pandemic	N1 > N2 hospitalization stay (median of 7 *versus* 5 days, p < 0.01) 39/41 patients required GC 33/39 patients with GC had a 79% clinical improvement during 24-h
Lee 2021 [38]	Retrospective study N = 8 teenagers who experienced EVALI and were followed after stopping the vaping and e-cigarette use	Average follow-up of 46.5 days All patients (8/8) received GC amid EVALI Radiography and spirometry parameters improved post-discontinuation of GC
Kaous 2020 [39]	Retrospective study N = 8 teenagers who experienced EVALI and received GC	2/8 patients: no GC when discharged 6/8 patients: GC when discharged (20–40 mg/day, 2–4 weeks)
Corcoran 2020 [40]	Retrospective study N = 7 teenagers with EVALI	42% of the patients received systemic steroids (as inpatients)
Carroll 2020 [43]	Retrospective study N = 15 teenagers with EVALI	GC use offered a clinical improvement 5/11 patients: inhaled GC (as outpatients)
Kass 2020 [41]	Retrospective study N = 10 teenagers with lung illness (8/10 patients with EVALI)	8/10 patients with EVALI (requiring prednisone, methyl prednisolone from 4 days to 8 weeks)
Fryman 2020 [44]	Retrospective study N = 8 adults (5 males and 3 females) with a median age of 31.5 (range: 24–62) years	50% (4/8) of patients required GC All patients had a clinical improvement Median hospitalization time: 7.5 (4–19) days
Rao 2020 [45]	Retrospective study N = 13 teenagers (54% were females)	12/13: required GC: 10/12: oral and intravenous GC 1/12: oral GC alone 1/12: intravenous glucocorticoids alone 11/12: clinical improvement (forced expiratory volume and forced vital capacity) following GC Median hospitalization time: 7 (2–120) days
Zou 2020 [46]	Retrospective study N = 36 individuals with median of 21 (19–30.5) years	72%: required GC (median daily prednisone equivalent: 46 mg)

Abbreviations: EVALI = e-cigarette and vaping-associated lung injury; GC = glucocorticoids.

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
