# Peer review of "Metabolic and Other Endocrine Elements with Regard to Lifestyle Choices: Focus on E-Cigarettes"

_metabolites, 2023, doi:10.3390/metabo13121192_

Round 1

Reviewer 1 Report

Comments and Suggestions for Authors

Line 46: “Original studies” is vague. Specify if it is experimental or primary studies. Also, before you begin categorizing studies, state how many total studies were included in the narrative review

Line 53: Again, be specific if it type-1 or type-2 diabetes.

Line 58: What is meant by “attainment of obesity”?

Line 90-91: The sentence “different levels of statistical evidence so far” does not make sense. Are you trying to say that there is no consensus if e-cigs are better or worse than traditional cigarettes? There needs to more clarity.

Lines 105-111: This paragraph does not make sense. It makes it seem as if Covid-19 will be looked at under the review but it is not. It is like a red herring. For the introduction, try to only mention topics that will be included into the review.

Line 116: Specify if it is primary or experimental studies.

Figure 1: The authors should look at the PRISMA diagram (http://www.prisma-statement.org/?AspxAutoDetectCookieSupport=1) when reporting the results of the search. This is a standard that is used by many reviews. The authors also look at the other recommendations given by the Cochrane reviews such as the number of reviewers that conducted the search, that screened the papers, etc..

Lines 131-137: Although the paragraph is appreciated, this is not the place for it. It should be moved to the discussion. The “Results” is only for the reporting of findings, not implications or rationale.

Line 155: Refrain from referring to studies as “larger study”. Be specific when talking about the studies.

Line 177: Describe which type of diabetes the studies talk about as there may be different implications.

Line 183: Again refrain from referring to studies as “larger study”.

Line 187: The term “respectively” should come after the second term being described.

Line 270: Similar to the other paragraphs, include a sentence that describes how many studies were found that talk about metabolic syndrome and e-cigs

Discussion: The discussion would be better to understand if it follows the same description as was used in the results. That is, the Discussion talks about prediabetes, followed by Diabetes mellitus, etc..

Comments on the Quality of English Language

The use of the English language is very good. Some minor revisions but otherwise, it is very good. 

Author Response

Response to Review 1 Comments

Dear Reviewer,

Thank you very much for your time and your effort to review our manuscript.

We are very grateful for providing your valuable feedback on the article.

Here is our response and related amendment that has been made in the manuscript according to your review (marked in yellow color).

Line 46: “Original studies” is vague. Specify if it is experimental or primary studies. Also, before you begin categorizing studies, state how many total studies were included in the narrative review

Thank you very much. We specified these data according to your recommendations.

“We included original (clinical) studies and excluded case reports, reviews, and editorials, and experimental studies.”

“……..(n=22). (Figure 1)….”

Thank you.

Line 53: Again, be specific if it type-1 or type-2 diabetes.

Thank you very much. We specified this important aspect at Results section, as well as Abstract.

“A total of three studies addressed the issue of type 2 diabetes and e-cig. Of note, all the data we found only included this type of diabetes which is a typical component in patients with various metabolic anomalies.”

Thank you.

Line 58: What is meant by “attainment of obesity”?

Thank you very much. We corrected it (“assessment”). Thank you.

Line 90-91: The sentence “different levels of statistical evidence so far” does not make sense. Are you trying to say that there is no consensus if e-cigs are better or worse than traditional cigarettes? There needs to more clarity.

Thank you very much. “Different levels of statistical evidence” (https://libguides.winona.edu/ebptoolkit/Levels-Evidence) means that some data are supported by less powerful studies (from a statistical perspective), other data are still incompletely supported by well-designed studies, while other aspects are supported by meta-analysis (particularly involving respiratory and cardiovascular issues in e-cig users).

Moreover, these various studies (underlying “different levels of statistical evidence”) did not support the idea of being completely safe, thus, while still representing an alternative to the traditional smoking (and well known side effects), e-cig are not completely harmless as far as we know at this moment.

Thank you.

Lines 105-111: This paragraph does not make sense. It makes it seem as if Covid-19 will be looked at under the review but it is not. It is like a red herring. For the introduction, try to only mention topics that will be included into the review.

Thank you. We mentioned the recent COVID-19 pandemic for the following aspects:

  1. Recent data showed an increase of e-cig use amid COVID-19 pandemic regulations (such as isolation, outdoor activities restrictions, reduced of social activities, etc.)
  2. During COVID-19 pandemic, the presentation at emergency room, especially for young patients, required a rapid differential diagnosis between a severe viral infection and EVALI or VALI.
  3. Considering these general medical and social aspects that we briefly mentioned, we respectfully consider that it is essential to frame our research with respect to recent pandemic in order to highlight the importance of addressing this topic despite the fact that we choose an endocrine and metabolic perspective.

Thank you.

Line 116: Specify if it is primary or experimental studies.

Thank you very much. We already addressed this point. Thank you.

Figure 1: The authors should look at the PRISMA diagram (http://www.prisma-statement.org/?AspxAutoDetectCookieSupport=1) when reporting the results of the search. This is a standard that is used by many reviews. The authors also look at the other recommendations given by the Cochrane reviews such as the number of reviewers that conducted the search, that screened the papers, etc..

Thank you very much. PRISMA stands for “Preferred Reporting Items for Systematic Reviews and Meta-Analyses” (http://prisma-statement.org/prismastatement/flowdiagram.aspx) and Cochrane Review is a systematic review of research in health (https://www.cochranelibrary.com/about/about-cochrane-reviews) and this is a narrative review, not a systematic one. Due to the heterogeneity of the spectrum of e-cig and endocrine/metabolic aspects, we choose to introduce the data as a narrative review since various levels of statistical evidence are identified in mentioned papers. On the other hand, a systematic review pinpoints a specific critical assessment which in this particular matter remains rather limited so far.

However, this type of review (namely, narrative) is a well-recognized, standard, traditional approach which is suitable for topics with less generous publications so far such as an endocrine update of e-cig. This allowed us to examine and evaluate the scientific panel on this specific topic in a useful way for various practitioners. 

We mentioned at Discussion the aspect of this research and this paper approach.

“As limits of the current work we mention the narrative review type”

With respect to each author’ contribution to the research for this paper, the specific MDPI section at the end of the article included the authors’ initials and their work.

Thank you.

Lines 131-137: Although the paragraph is appreciated, this is not the place for it. It should be moved to the discussion. The “Results” is only for the reporting of findings, not implications or rationale.

Thank you very much. We moved it according to your suggestion. Thank you.

Line 155: Refrain from referring to studies as “larger study”. Be specific when talking about the studies.

Thank you very much. We correct it. Thank you.

Line 177: Describe which type of diabetes the studies talk about as there may be different implications.

Thank you very much. We researched “diabetes”, but the only data we could find were on type 2 diabetes and we prior mentioned this aspect. Thank you.

Line 183: Again refrain from referring to studies as “larger study”.

Thank you very much. We correct it. Thank you.

Line 187: The term “respectively” should come after the second term being described.

Thank you very much. We correct it. Thank you.

Line 270: Similar to the other paragraphs, include a sentence that describes how many studies were found that talk about metabolic syndrome and e-cigs.

Thank you very much. We specified the number.

“3.1.4. Metabolic syndrome

Among the two studies addressing the metabolic syndrome in e-cig,…”

Thank you.

Discussion: The discussion would be better to understand if it follows the same description as was used in the results. That is, the Discussion talks about prediabetes, followed by Diabetes mellitus, etc..

Thank you very much. We introduced the following sections at Discussion:

4.1. E-cig: is it time for taking into consideration a new of endocrine disruptor?

4.2. E-cig and data with regard to prediabetes and diabetes

4.3. E-cig users and obesity

4.3. Metabolic syndrome among e-cig users

4.4. Fertility profile regarding e-cig

4.5. Thyroid assessment in subjects using e-cig

4.6. Bone status, including peak bone mass, in teenagers and young adults using e-cig

4.7. Cannabinoid receptors

4.8. Further expansion of e-cig – related endocrine researches

Thank you.

The use of the English language is very good. Some minor revisions but otherwise, it is very good. 

Thank you very much. We reviewed the language. Thank you.

Thank you very much.

Reviewer 2 Report

Comments and Suggestions for Authors

Although it is a review article, the authors should drive the conclusion with cited studies to provide current reports on the associations between e-cigarettes and endocrine diseases. The abstract was too long and the contents/expressions in the tables were not unified. Most of the text was written quite complicated. Thus, it is not easy to follow the manuscript and difficult to conclude the effects of e-cigarettes on endocrine diseases.

Minor: N3 and N3 in the study design of Kim 2022, Table 1,  should be separated in different lines. 

Comments on the Quality of English Language

Lines 78-77 should be separated. I doubt the authors missed the full stop (.) after".,,, determined'

Author Response

Response to Review 2 Comments

Dear Reviewer,

Thank you very much for your time and your effort to review our manuscript.

We are very grateful for your insightful comments and observations, also, for providing your valuable feedback on the article.

Here is a point-by-point response and related amendments that have been made in the manuscript according to your review (marked in yellow color).

Although it is a review article, the authors should drive the conclusion with cited studies to provide current reports on the associations between e-cigarettes and endocrine diseases.

Thank you. We expanded the Conclusion section according to your recommendation. Thank you

“Overall, the most important metabolic highlights within the mentioned studies showed that among dual smokers there is a 1.57-fold prediabetes increase compare to those never smoking, with an increased prevalence of 10.2% (dual) versus 5.9% (e-cig-only). Current non-smokers who have been dual smokers still have an increased odd of prediabetes (a 1.27-fold risk increase). A lower prevalence of type 2 diabetes among dual users (3.3%) versus cigarette smoking (5.9%) was identified. One study found that e-cig use was associated with obesity in general population; another showed that obesity had a higher prevalence in dual smokers (51%) versus cig-only (41.2%, p<0.05), while another found that females (not males) e-cig smokers had higher body mass index. Data on metabolic syndrome are provided for dual smokers: one case-control study found that female dual smoking (not males) had higher odds of this syndrome than non-smokers. The need for awareness with respect to potential e-cigarette – associated medical issues should be part of modern medicine, including daily anamnesis. Whether metabolic/endocrine frame is part of this general picture is yet to be determined Surveillance protocols should help the clinicians to easily access the medical background of one subject, including this specific matter of electronic cigarettes with/without conventional cigarettes smoking and other habits/lifestyle elements, especially when taken into consideration metabolism anomalies.”

The abstract was too long and the contents/expressions in the tables were not unified.

Thank you. We reduced the abstract and corrected it.

“Our objective was to overview recent data on metabolic/endocrine disorders with respect to e-cigarette use. This is a narrative review; we researched English, full-length original articles on PubMed (between January 2020 and August 2023) by using different keywords in the area of metabolic/endocrine issues. We only included original clinical studies (n=22) and excluded case reports, and experimental studies. 3 studies (N1=22,385; N2=600,046; N3=5101) addressed prediabetes risk; N1 showed a 1.57-fold increase risk of dual vs. never smokers, a higher risk that was not confirmed in N2 study (based on self-reported assessments). Current non-smokers (N1) who have been dual smokers still have an increased odd of prediabetes (a 1.27-fold risk increase). N3 and another 2 studies addressed type 2 diabetes mellitus (DM): a lower prevalence of DM among dual users (3.3%) vs. cigarette smoking (5.9%) was identified. 6 studies investigated obesity profile (4 of them found certain correlations with e-cig use). One study (N4=373,781) showed that e-cig use was associated with obesity in the general population (OR=1.6, 95%CI:1.3–2.1, p=0.0007); another (N5=7505, 0.82% were e-cig-only) showed that obesity had a higher prevalence in dual smokers (51%) vs. cig-only (41.2%, p<0.05), while another (N6=3055) found that females (not males) e-cig smokers had higher body mass index vs. non-smokers. Data on metabolic syndrome (MS) are provided for dual smokers (n=2): one case-control study found that female dual smoking (not males) had higher odds of MS than non-smokers. The need for awareness with respect to potential e-cig – associated medical issues should be part of modern medicine, including daily anamnesis. Whether metabolic/endocrine frame is part of the general picture is yet to be determined. Surveillance protocols should help the clinicians to easily access the medical background of one subject, including this specific matter of e-cig with/without conventional cigarettes smoking and other habits/lifestyle elements, especially when taken into consideration metabolism anomalies”

Thank you

Most of the text was written quite complicated. Thus, it is not easy to follow the manuscript and difficult to conclude the effects of e-cigarettes on endocrine diseases.

Thank you. We introduced new subsections at Results and Discussion section according to your recommendations.

We introduced the sections 3.2.1. (3.2.1.A. and 3.2.1.B) and 3.2.2. for Results section, as following:

3.2.1. Potential fertility issues

3.2.1. A. Female fertility

3.2.1. B. Male fertility

3.2.2. EVALI and the use of systemic steroids

Moreover, we introduced the following sections at Discussion:

4.1. E-cig: is it time for taking into consideration a new of endocrine disruptor?

4.2. E-cig and data with regard to prediabetes and diabetes

4.3. E-cig users and obesity

4.3. Metabolic syndrome among e-cig users

4.4. Fertility profile regarding e-cig

4.5. Thyroid assessment in subjects using e-cig

4.6. Bone status, including peak bone mass, in teenagers and young adults using e-cig

4.7. Cannabinoid receptors

4.8. Further expansion of e-cig – related endocrine researches

Thank you. 

Minor: N3 and N3 in the study design of Kim 2022, Table 1, should be separated in different lines. 

Thank you very much. We correct it. Thank you.

Lines 78-77 should be separated. I doubt the authors missed the full stop (.) after".,,, determined'

Thank you very much. We correct it. Thank you.

Reviewer 3 Report

Comments and Suggestions for Authors

The manuscript's subject is interesting, as e-cigarettes have gained a great popularity, especially among young people, and their health impact is not yet fully understood. 

I have some observations/suggestions for the authors:

- I noticed some language/typing mistakes;

- The abstract section is too long; the authors should reorganize the text and emphasize the main ideas;

- Line 93 - metabolic interferences, particularly in patients with increased body mass index (BMI) were identified – reference needed;

- I suggest adding some extra information in the diagram presenting the design of the study (for example, the records removed before screening are duplicates?; give examples of some exclusion criteria other than the duplicates);

- Line 160 - One factor that might have influenced the difference in prevalence is the less precise method of prediabetes diagnosis used – please elaborate;

- I found another study about the connection between e-cigarettes and combustible cigarettes and prediabetes, diabetes, or insulin resistance that could be included for discussion in the present paper;

Jiahui Cai, Aurelian Bidulescu, The association between e-cigarette use or dual use of e-cigarette and combustible cigarette and prediabetes, diabetes, or insulin resistance: Findings from the National Health and Nutrition Examination Survey (NHANES), Drug and Alcohol Dependence, Volume 251, 2023, 110948, ISSN 0376-8716, https://doi.org/10.1016/j.drugalcdep.2023.110948. (https://www.sciencedirect.com/science/article/pii/S0376871623011869).

- In the Discussion section, the data could be arranged respecting the same order as in the Results section – diabetes, obesity, metabolic syndrome, fertility…, and, also, the addition of subtitles could be useful.

Author Response

Response to Review 3 Comments

Dear Reviewer,

Thank you very much for your time and your effort to review our manuscript.

We are very grateful for your insightful comments and observations, also, for providing your valuable feedback on the article.

Here is a point-by-point response and related amendments that have been made in the manuscript according to your review (marked in yellow color).

The manuscript's subject is interesting, as e-cigarettes have gained a great popularity, especially among young people, and their health impact is not yet fully understood. 

Thank you very much. We really appreciate it!

I have some observations/suggestions for the authors:

Thank you very much. We followed each of them. Thank you.

I noticed some language/typing mistakes;

Thank you very much. We revisited the text. Thank you

The abstract section is too long; the authors should reorganize the text and emphasize the main ideas.

Thank you very much. We reduced the length of the abstract and reorganized the text.

 “Our objective was to overview recent data on metabolic/endocrine disorders with respect to e-cigarette use. This is a narrative review; we researched English, full-length original articles on PubMed (between January 2020 and August 2023) by using different keywords in the area of metabolic/endocrine issues. We only included original clinical studies (n=22) and excluded case reports, and experimental studies. 3 studies (N1=22,385; N2=600,046; N3=5101) addressed prediabetes risk; N1 showed a 1.57-fold increase risk of dual vs. never smokers, a higher risk that was not confirmed in N2 study (based on self-reported assessments). Current non-smokers (N1) who have been dual smokers still have an increased odd of prediabetes (a 1.27-fold risk increase). N3 and another 2 studies addressed type 2 diabetes mellitus (DM): a lower prevalence of DM among dual users (3.3%) vs. cigarette smoking (5.9%) was identified. 6 studies investigated obesity profile (4 of them found certain correlations with e-cig use). One study (N4=373,781) showed that e-cig use was associated with obesity in the general population (OR=1.6, 95%CI:1.3–2.1, p=0.0007); another (N5=7505, 0.82% were e-cig-only) showed that obesity had a higher prevalence in dual smokers (51%) vs. cig-only (41.2%, p<0.05), while another (N6=3055) found that females (not males) e-cig smokers had higher body mass index vs. non-smokers. Data on metabolic syndrome (MS) are provided for dual smokers (n=2): one case-control study found that female dual smoking (not males) had higher odds of MS than non-smokers. The need for awareness with respect to potential e-cig – associated medical issues should be part of modern medicine, including daily anamnesis. Whether metabolic/endocrine frame is part of the general picture is yet to be determined. Surveillance protocols should help the clinicians to easily access the medical background of one subject, including this specific matter of e-cig with/without conventional cigarettes smoking and other habits/lifestyle elements, especially when taken into consideration metabolism anomalies”

We introduced the sections 3.2.1. (3.2.1.A. and 3.2.1.B) and 3.2.2. for Results section, as following:

3.2.1. Potential fertility issues

3.2.1. A. Female fertility

3.2.1. B. Male fertility

3.2.2. EVALI and the use of systemic steroids

Moreover, we introduced new sections at Discussion, as following:

4.1. E-cig: is it time for taking into consideration a new of endocrine disruptor?

4.2. E-cig and data with regard to prediabetes and diabetes

4.3. E-cig users and obesity

4.3. Metabolic syndrome among e-cig users

4.4. Fertility profile regarding e-cig

4.5. Thyroid assessment in subjects using e-cig

4.6. Bone status, including peak bone mass, in teenagers and young adults using e-cig

4.7. Cannabinoid receptors

4.8. Further expansion of e-cig – related endocrine researches

Thank you

Line 93 - metabolic interferences, particularly in patients with increased body mass index (BMI) were identified – reference needed;

Thank you very much. We correct it. Thank you

I suggest adding some extra information in the diagram presenting the design of the study (for example, the records removed before screening are duplicates?; give examples of some exclusion criteria other than the duplicates).

Thank you very much. We corrected and expanded the flowchart diagram. Thank you.

Line 160 - One factor that might have influenced the difference in prevalence is the less precise method of prediabetes diagnosis used – please elaborate.

Thank you very much. We provided more details with respect to the mentioned statement.

“One factor that might have influenced the difference in prevalence is the less precise method of prediabetes diagnosis used by Zhang et al. [14], meaning the fact that the prevalence of prediabetes was established upon asking the subjects if a doctor previously informed each of them as having the diagnosis of “prediabetes” or “borderline diabetes” [14].

Thank you

I found another study about the connection between e-cigarettes and combustible cigarettes and prediabetes, diabetes, or insulin resistance that could be included for discussion in the present paper: Jiahui Cai, Aurelian Bidulescu, The association between e-cigarette use or dual use of e-cigarette and combustible cigarette and prediabetes, diabetes, or insulin resistance: Findings from the National Health and Nutrition Examination Survey (NHANES), Drug and Alcohol Dependence, Volume 251, 2023, 110948, ISSN 0376-8716, https://doi.org/10.1016/j.drugalcdep.2023.110948. (https://www.sciencedirect.com/science/article/pii/S0376871623011869).

Thank you very much. We introduced this study, as well, according to your recommendation.

We introduced the mentioned reference accordingly:

Cai J, Bidulescu A. The association between e-cigarette use or dual use of e-cigarette and combustible cigarette and prediabetes, diabetes, or insulin resistance: Findings from the National Health and Nutrition Examination Survey (NHANES).Drug Alcohol Depend. 2023;251:110948. doi:10.1016/j.drugalcdep.2023.110948. 

We updated the tables at prediabetes and diabetes sections.

First author

Year of publication

Reference

number

Study design and population

Key findings

Prediabetes

Cai

2023

[5]

Cross-sectional study (NHANES) *

N=5101 adults from U.S.

N1=6.3% current e-cig users

N2=17.1% former e-cig users

E-cig use was not correlated with prediabetes (p>0.05) – fully adjusted model

No correlation between dual use and prediabetes (p>0.05)

Diabetes mellitus

Cai

2023

[5]

*

E-cig use was not correlated with diabetes (p>0.05) – fully adjusted model

No correlation between dual use and diabetes (p>0.05)

No correlation between dual use and insulin resistance (p>0.05)

N1 were 63% (95%CI:1-2.91) more likely to have higher HOMA-IR than never e-cig users

N2 were 64% (95%CI:1.04-2.59) more likely to have higher HOMA-IR than never e-cig users

We also extended the data from Results, Discussion and Abstract sections according to this new input, for instance:

 “According to our methods, we identified three studies to address the issue of e-cigarettes and prediabetes, all of them having a group with dual e-cigarettes users and two included a specific subgroup with e-cigarette only smoking [5,13,14]”

“On the other hand, the third study conducted by Cai et al. [5] did not find a clear correlation between e-cig and prediabetes [5].”

“As mentioned the study of Cai et al. looked at the diabetes risk, as well; an e-cig used were more likely to present a increased index of insulin resistance (particularly, HOMA-IR) than the persons who were never e-cig users [5].

“…..probably the assessment of insulin resistance might provide additional clues of diabetes risk”..

In the Discussion section, the data could be arranged respecting the same order as in the Results section – diabetes, obesity, metabolic syndrome, fertility…, and, also, the addition of subtitles could be useful.

Thank you very much. We introduced these sections at Discussion chapter:

4.1. E-cig: is it time for taking into consideration a new of endocrine disruptor?

4.2. E-cig and data with regard to prediabetes and diabetes

4.3. E-cig users and obesity

4.3. Metabolic syndrome among e-cig users

4.4. Fertility profile regarding e-cig

4.5. Thyroid assessment in subjects using e-cig

4.6. Bone status, including peak bone mass, in teenagers and young adults using e-cig

4.7. Cannabinoid receptors

4.8. Further expansion of e-cig – related endocrine researches

Thank you.

Round 2

Reviewer 1 Report

Comments and Suggestions for Authors

Line 52: What does “certain correlations” mean? Just say correlations or provide the exact direction (i.e. positive or negative or no)

Line 53+54: Try to stick to the same reporting convention for p-value. If you give the exact p-value, do that for all the studies mentioned. Or if you only give p<0.05, then stick to that. Otherwise, it may confuse the reader why you keep switching between them.

Line 85: You can say “unexpected negative effects on health”

Methods: It would good if the authors can provide a rationale as to why they chose to only look at the dates of January 2020 to August 2023. It may seem obvious because these are the years when Covid occurred but 10 years from now, that information may be not be as readily known or remembered.

Figure 1: The calculations for the studies included is wrong. When you subtract “Records screened” from “Records excluded”, you get 2, not 22. The authors should go back and review their notes to see where the missing 4 papers are from or check their values again.

Lines 461-470: It makes more sense to move these sentences to section 4.4 where the effects of e-cig and fertility are talked about at.

Section 4.7- I am confused as to the point of the paragraph. It starts out with the cannabinoid receptors but then ventures into immune responses without ever going back to cannabinoid receptors.

Author Response

Response to Review 1 Comments – Second Round

Dear Reviewer,

Thank you very much for your time and your effort to review our manuscript for the second round.

We are very grateful for providing your valuable feedback on the article.

Here is our response and related amendment that has been made in the manuscript according to your review (marked in green color).

Line 52: What does “certain correlations” mean? Just say correlations or provide the exact direction (i.e. positive or negative or no).

Thank you very much. We corrected it (“positive”). Thank you

Line 53+54: Try to stick to the same reporting convention for p-value. If you give the exact p-value, do that for all the studies mentioned. Or if you only give p<0.05, then stick to that. Otherwise, it may confuse the reader why you keep switching between them.

Thank you very much. We corrected it (“p<0.05”). Thank you

Line 85: You can say “unexpected negative effects on health”.

Thank you very much. We corrected it based on your recommendation. Thank you

Methods: It would good if the authors can provide a rationale as to why they chose to only look at the dates of January 2020 to August 2023. It may seem obvious because these are the years when Covid occurred but 10 years from now, that information may be not be as readily known or remembered.

Thank you very much. We added this information according to your recommendation.

“The rational of choosing this specific time frame is related to the recent COVID-19 pandemic and associated regulations such as isolation, outdoor activities restrictions, reduced of social activities that seemed to be associated with an increase use of e-cig; also, the presentation at emergency room amid pandemic, especially for young patients, required a rapid differential diagnosis between a severe viral infection and EVALI or VALI [7].”

Thank you

Figure 1: The calculations for the studies included is wrong. When you subtract “Records screened” from “Records excluded”, you get 2, not 22. The authors should go back and review their notes to see where the missing 4 papers are from or check their values again.

Thank you very much. We corrected it (“51” instead of “47”). Thank you

Lines 461-470: It makes more sense to move these sentences to section 4.4 where the effects of e-cig and fertility are talked about at.

Thank you very much. We moved that specific lines to the section 4.4. according to your suggestion. Thank you

Section 4.7- I am confused as to the point of the paragraph. It starts out with the cannabinoid receptors but then ventures into immune responses without ever going back to cannabinoid receptors.

Thank you very much. This aspect was mentioned as another topic that is less understood in this specific matter of e-cig use and potential negative side effects with respect to the metabolic and endocrine aspects. We corrected the section by introducing the followings:

“Whether the crossroads with cannabinoid system and catecholamines status are essential players in metabolic and endocrine side effects of e-cig is still a matter of further studies. Other aspects are yet to be understood. For instance, we mention an interesting aspect that is also less clear nowadays: an acute hypersensitivity reaction was suspected to inhale antigen on a 16-year-old female who was confirmed with idiopathic acute eosinophilic pneumonia after she used e-cigarettes for 1 year (tetrahydrocannabinol cartridge) thus suggesting potential (antigen – induced) autoimmune effects [99].

Thank you

Thank you very much.

Reviewer 3 Report

Comments and Suggestions for Authors

The authors did a good job revising the manuscript. I have no further comments or questions.

Author Response

Response to Review 3 Comments

Dear Reviewer,

Thank you very much for your time and your effort to review our manuscript.

We are very grateful for your insightful comments and observations, also, for providing your valuable feedback on the article.

The authors did a good job revising the manuscript. I have no further comments or questions.

Thank you very much. We really appreciate it!

Round 3

Reviewer 1 Report

Comments and Suggestions for Authors

If possible, the authors should provide a list of the included and excluded studies in an Excel file.

Narrative reviews are very important as many may read them to understand the direction a field is heading in and is generally seen as a summary of recent findings in that field. Being able to see what studies were excluded and why would be great to show scientific rigor and transparency. 

Author Response

Review report – reviewer number 2 – third round

If possible, the authors should provide a list of the included and excluded studies in an Excel file.

Narrative reviews are very important as many may read them to understand the direction a field is heading in and is generally seen as a summary of recent findings in that field. Being able to see what studies were excluded and why would be great to show scientific rigor and transparency. 

Dear Reviewer number 2,

Thank you for the third round of review.

We introduced the list as followings  – the first 22 lines represent the included articles as specified in main text and tables of the original text (in red); the other lines up to the line number 192 represent the excluded papers.

This is an Excel list as you requested, however, the MDPI format allows us only a word or .pdf file – we attach this prove which is a Print Screen of the system.

We attach the list.

Best regards,

The authors

Cai J, Bidulescu A. The association between e-cigarette use or dual use of e-cigarette and combustible cigarette and prediabetes, diabetes, or insulin resistance: Findings from the National Health and Nutrition Examination Survey (NHANES).Drug Alcohol Depend. 2023;251:110948. doi:10.1016/j.drugalcdep.2023.110948

Kim SH, Park M, Kim GR, Joo HJ, Jang SI. Association of Mixed Use of Electronic and Conventional Cigarettes and Exposure to Secondhand Smoke With Prediabetes. J Clin Endocrinol Metab. 2022;107(1):e44-e56. doi:10.1210/clinem/dgab558.

Zhang Z, Jiao Z, Blaha MJ, Osei A, Sidhaye V, Ramanathan M Jr, Biswal S. The Association Between E-Cigarette Use and Prediabetes: Results From the Behavioral Risk Factor Surveillance System, 2016-2018. Am J Prev Med. 2022;62(6):872-877. doi:10.1016/j.amepre.2021.12.009.

Leavens ELS, Ford BR, Ojo-Fati O, Winkelman TNA, Vickery KD, Japuntich SJ, Busch AM. Electronic cigarette use patterns and chronic health conditions among people experiencing homelessness in MN: a statewide survey. BMC Public Health. 2020;20(1):1889. doi:10.1186/s12889-020-09919-4.

Zhao Z, Zhang M, Wu J, Xu X, Yin P, Huang Z, Zhang X, Zhou Y, Zhang X, Li C, Wang L, Gao GF, Wang L, Li X, Zhou M. E-cigarette use among adults in China: findings from repeated cross-sectional surveys in 2015-16 and 2018-19. Lancet Public Health. 2020;5(12):e639-e649. doi:10.1016/S2468-2667(20)30145-6.

Sompa SI, Zettergren A, Ekström S, Upadhyay S, Ganguly K, Georgelis A, Ljungman P, Pershagen G, Kull I, Melén E, Palmberg L, Bergström A. Predictors of electronic cigarette use and its association with respiratory health and obesity in young adulthood in Sweden; findings from the population-based birth cohort BAMSE. Environ Res. 2022;208:112760. doi:10.1016/j.envres.2022.112760.

Hoover LV, Yu HP, Cummings JR, Ferguson SG, Gearhardt AN. Co-occurrence of food addiction, obesity, problematic substance use, and parental history of problematic alcohol use. Psychol Addict Behav. 2022. doi:10.1037/adb0000870.

Alqahtani MM, Alanazi AMM, Almutairi AS, Pavela G. Electronic cigarette use is negatively associated with body mass index: An observational study of electronic medical records. Obes Sci Pract. 2020;7(2):226-231. doi:10.1002/osp4.468.

Oh SS, Jang JE, Lee DW, Park EC, Jang SI. Cigarette type or smoking history: Which has a greater impact on the metabolic syndrome and its components? Sci Rep. 2020;10(1):10467. doi:10.1038/s41598-020-67524-2.

Galanti F, Licata E, Paciotti G, Gallo M, Riccio S, Miriello D, Dal Lago A, Meneghini C, Fabiani C, Antonaci D, Schiavi MC, Scudo M, Salacone P, Sebastianelli A, Battaglia FA, Rago R. Impact of different typologies of smoking on ovarian reserve and oocyte quality in women performing ICSI cycles: an observational prospective study. Eur Rev Med Pharmacol Sci. 2023;27(11):5190-5199. doi:10.26355/eurrev_202306_32637.

Harlow AF, Hatch EE, Wesselink AK, Rothman KJ, Wise LA. Electronic Cigarettes and Fecundability: Results From a Prospective Preconception Cohort Study. Am J Epidemiol. 2021;190(3):353-361. doi:10.1093/aje/kwaa067.

Holmboe SA, Priskorn L, Jensen TK, Skakkebaek NE, Andersson AM, Jørgensen N. Use of e-cigarettes associated with lower sperm counts in a cross-sectional study of young men from the general population. Hum Reprod. 2020;35(7):1693-1701. doi:10.1093/humrep/deaa089.

Abdallah B, Lee H, Weerakoon SM, Messiah SE, Harrell MB, Rao DR. Clinical manifestations of EVALI in adolescents before and during the COVID-19 pandemic.  Pediatr Pulmonol. 2023;58(3):949-958. doi:10.1002/ppul.26283. 

Lee SA, Sayad E, Yenduri NJS, Wang KY, Guillerman RP, Farber HJ. Improvement in Pulmonary Function Following Discontinuation of Vaping or E-Cigarette Use in Adolescents with EVALI. Pediatr Allergy Immunol Pulmonol. 2021;34(1):23-29. doi:10.1089/ped.2020.1270.

Kaous M, Xian J, Rongo D, McDonald M, Ocasionez D, Mathew R, Estrada-Y-Martin RM, Patel B, Cherian SV, Jani PP. Clinical, radiology, pathologic patterns and outcomes of vaping related pulmonary injury in a single institution; A case series. Respir Med. 2020;173:106153. doi:10.1016/j.rmed.2020.106153.

Corcoran A, Carl JC, Rezaee F. The importance of anti-vaping vigilance-EVALI in seven adolescent pediatric patients in Northeast Ohio. Pediatr Pulmonol. 2020;55(7):1719-1724. doi:10.1002/ppul.24872. 

Kass AP, Overbeek DL, Chiel LE, Boyer EW, Casey AMH. Case series: Adolescent victims of the vaping public health crisis with pulmonary complications. Pediatr Pulmonol. 2020;55(5):1224-1236. doi:10.1002/ppul.24729.

Carroll BJ, Kim M, Hemyari A, Thakrar P, Kump TE, Wade T, De Vela G, Hall J, Diaz CD, D'Andrea LA. Impaired lung function following e-cigarette or vaping product use associated lung injury in the first cohort of hospitalized adolescents. Pediatr Pulmonol. 2020;55(7):1712-1718. doi:10.1002/ppul.24787. 

Fryman C, Lou B, Weber AG, Steinberg HN, Khanijo S, Iakovou A, Makaryus MR. Acute Respiratory Failure Associated With Vaping. Chest. 2020;157(3):e63-e68. doi:10.1016/j.chest.2019.10.057.

Rao DR, Maple KL, Dettori A, Afolabi F, Francis JKR, Artunduaga M, Lieu TJ, Aldy K, Cao DJ, Hsu S, Feng SY, Mittal V. Clinical Features of E-cigarette, or Vaping, Product Use-Associated Lung Injury in Teenagers. Pediatrics. 2020;146(1):e20194104. doi:10.1542/peds.2019-4104.

Zou RH, Tiberio PJ, Triantafyllou GA, Lamberty PE, Lynch MJ, Kreit JW, McVerry BJ, 3. Gladwin MT, Morris A, Chiarchiaro J, Fitzpatrick ME, Rose JJ. Clinical Characterization of E-Cigarette, or Vaping, Product Use-associated Lung Injury in 36 Patients in Pittsburgh, Pennsylvania. Am J Respir Crit Care Med. 2020;201(10):1303-1306. doi:10.1164/rccm.202001-0079LE.

Kim CY, Paek YJ, Seo HG, Cheong YS, Lee CM, Park SM, Park DW, Lee K. Dual use of electronic and conventional cigarettes is associated with higher cardiovascular risk factors in Korean men. Sci Rep. 2020;10(1):5612. doi:10.1038/s41598-020-62545-3.

Awad K, Mohammed M, Martin SS, Banach M. Association between electronic nicotine delivery systems use and risk of stroke: a meta-analysis of 1,024,401 participants. Arch Med Sci. 2023;19(5):1538-1540. doi:10.5114/aoms/171473.

Kotewar SS, Pakhale A, Tiwari R, Reche A, Singi SR. Electronic Nicotine Delivery System: End to Smoking or Just a New Fancy Cigarette. Cureus. 2023;15(8):e43425. doi:10.7759/cureus.43425.

Mravec B, Tibensky M, Horvathova L, Babal P. E-Cigarettes and Cancer Risk. Cancer Prev Res (Phila). 2020;13(2):137-144. doi:10.1158/1940-6207.CAPR-19-0346.

Belkin S, Benthien J, Axt PN, Mohr T, Mortensen K, Weckmann M, Drömann D, Franzen KF. Impact of Heated Tobacco Products, E-Cigarettes, and Cigarettes on Inflammation and Endothelial Dysfunction. Int J Mol Sci. 2023;24(11):9432. doi:10.3390/ijms24119432.

Abelia XA, Lesmana R, Goenawan H, Abdulah R, Barliana MI. Comparison impact of cigarettes and e-cigs as lung cancer risk inductor: a narrative review. Eur Rev Med Pharmacol Sci. 2023;27(13):6301-6318. doi:10.26355/eurrev_202307_32990.

Rose JJ, Krishnan-Sarin S, Exil VJ, Hamburg NM, Fetterman JL, Ichinose F, Perez-Pinzon MA, Rezk-Hanna M, Williamson E; American Heart Association Council on Cardiopulmonary, Critical Care, Perioperative and Resuscitation; Council on Epidemiology and Prevention; Council on Cardiovascular Radiology and Intervention; Council on Lifestyle and Cardiometabolic Health; Council on Peripheral Vascular Disease; Stroke Council; and Council on Arteriosclerosis, Thrombosis and Vascular Biology. Cardiopulmonary Impact of Electronic Cigarettes and Vaping Products: A Scientific Statement From the American Heart Association. Circulation. 2023. doi:10.1161/CIR.0000000000001160.

Boloña E, Felix M, Vanegas E, Vera Paz C, Cherrez-Ojeda I.  A Case of Vaping-associated Pulmonary Illness in South America: Highlighting the Need for Awareness and Surveillance Programs in the Region. Am J Respir Crit Care Med. 2020;201(6):733-735. doi:10.1164/rccm.201910-2002LE.

Jacobs W, Nabors L, Mahabee-Gittens ME, Merianos AL. E-cigarette and marijuana use and the attainment of obesity prevention guidelines among U.S. adolescents. Prev Med Rep. 2021;23:101445. doi:10.1016/j.pmedr.2021.101445.

Mason TB, Leventhal AM. Weight Status and Effects of Non-Tobacco Flavors on E-Cigarette Product Appeal. Subst Use Misuse. 2021;56(6):848-853. doi:10.1080/10826084.2021.1899229.

Mason TB, Leventhal AM. Relations among sweet taste preference, body mass index, and use of E-cigarettes for weight control motives in young adults. Eat Behav. 2021;41:101497. doi:10.1016/j.eatbeh.2021.101497.

Overbeek DL, Kass AP, Chiel LE, Boyer EW, Casey AMH. A review of toxic effects of electronic cigarettes/vaping in adolescents and young adults. Crit Rev Toxicol. 2020;50(6):531-538. doi:10.1080/10408444.2020.1794443.

 Kjeld SG, Andersen S, Andersen A, Glenstrup S, Lund L, Danielsen D, Bast LS. Who are the young users of tobacco products? Prevalence and characteristics of Danish adolescents who have either smoked cigarettes, used alternative tobacco products, or used both. Nordisk Alkohol Nark. 2021;38(6):555-572. doi:10.1177/14550725211027687.

Callahan SJ, Harris D, Collingridge DS, Guidry DW, Dean NC, Lanspa MJ, Blagev DP. Diagnosing EVALI in the Time of COVID-19. Chest. 2020;158(5):2034-2037. doi:10.1016/j.chest.2020.06.029. 

Saygın H, Korgalı E, Koç T, Doğan K. The effect of smoking and electronic cigarettes on rat testicles. Rev Int Androl. 2023;21(3):100365. doi:10.1016/j.androl.2023.100365

Petrache I, Gupta A, Hume PS, Rivera T, Schweitzer KS, Chu HW. Pathogenesis of E-Cigarette Vaping Product Use-Associated Lung Injury (EVALI). Compr Physiol. 2023;13(2):4617-4630. doi:10.1002/cphy.c220022.

Tattan-Birch H, Jackson S, Shahab L, Hartmann-Boyce J, Kock L, Simonavicius E et al. Heated tobacco products for smoking cessation and reducing smoking prevalence. Cochrane Database Syst Rev. 2020;11. https://doi.org/10.1002/14651858.CD013790.

Hartmann-Boyce J, McRobbie H, Butler AR, Lindson N, Bullen C, Begh R, Theodoulou A, Notley C, Rigotti NA, Turner T, Fanshawe TR, Hajek P. Electronic cigarettes for smoking cessation. Cochrane Database of Systematic Reviews. 2021;9. doi:10.1002/14651858.

Morean ME, Kong G, Bold KW, Davis DR, Krishnan-Sarin S. Accurately classifying cannabis blunt use as tobacco-cannabis co-use versus exclusive cannabis use. Drug Alcohol Depend. 2023;249:109941. doi:10.1016/j.drugalcdep.2023.109941.

Cao DJ, Aldy K, Hsu S, McGetrick M, Verbeck G, de Silva I, et al. Review of health consequences of electronic cigarettes and the outbreak of electronic cigarette, or vaping product use-associated lung injury. J Med toxicol. 2020;16:295–310.

Gale N, McEwan M, Camacho OM, Hardie G, Murphy J, Proctor CJ. Changes in Biomarkers of Exposure on Switching From a Conventional Cigarette to the glo Tobacco Heating Product: A Randomized, Controlled Ambulatory Study. Nicotine Tob. Res. 2021;23:584–591. doi:10.1093/ntr/ntaa135.

Szumilas K, Szumilas P, Grzywacz A, Wilk A. The Effects of E-Cigarette Vapor Components on the Morphology and Function of the Male and Female Reproductive Systems: A Systematic Review. Int J Environ Res Public Health. 2020;17(17):6152. doi:10.3390/ijerph17176152.

Barhdadi S, Courselle P, Deconinck E, Vanhee C. The analysis of cannabinoids in e-cigarette liquids using LC-HRAM-MS and LC-UV. J Pharm Biomed Anal. 2023;230:115394. doi:10.1016/j.jpba.2023.115394.

Montjean D, Godin Pagé MH, Bélanger MC, Benkhalifa M, Miron P. An Overview of E-Cigarette Impact on Reproductive Health. Life (Basel). 2023;13(3):827. doi:10.3390/life13030827.

Fadeyi O, Randhawa A, Shankar A, Garabetian C, Singh H, Topacio A. Thromboembolism Triggered by a Combination of Electronic Cigarettes and Oral Contraceptives: A Case Report and Review of Literature. J Investig Med High Impact Case Rep. 2023;11:23247096231181072. doi:10.1177/23247096231181072.

Vivarelli F, Granata S, Rullo L, Mussoni M, Candeletti S, Romualdi P, Fimognari C, Cruz-Chamorro I, Carrillo-Vico A, Paolini M, Canistro D. On the toxicity of e-cigarettes consumption: Focus on pathological cellular mechanisms. Pharmacol Res. 2022;182:106315. doi:10.1016/j.phrs.2022.106315.

Kechter A, Ceasar RC, Simpson KA, Schiff SJ, Dunton GF, Bluthenthal RN, Barrington-Trimis JL. A chocolate cake or a chocolate vape? Young adults describe their relationship with food and weight in the context of nicotine vaping. Appetite. 2022;175:106075. doi:10.1016/j.appet.2022.106075.

Ganson KT, Nagata JM. Associations between vaping and eating disorder diagnosis and risk among college students. Eat Behav. 2021;43:101566. doi:10.1016/j.eatbeh.2021.101566.

Mason TB, Tackett AP, Smith CE, Leventhal AM. Tobacco product use for weight control as an eating disorder behavior: Recommendations for future clinical and public health research. Int J Eat Disord. 2022;55(3):313-317. doi:10.1002/eat.23651.

Naveed A, Dang N, Gonzalez P, Choi SH, Mathew A, Wardle M, Garey L, Hamidovic A. E-Cigarette Dependence and Weight-Related Attitudes/Behaviors Associated With Eating Disorders in Adolescent Girls. Front Psychiatry. 2021;12:713094. doi:10.3389/fpsyt.2021.713094.

Dunn CP, Riley JB, Hawkins KB, Tercyak KP. Factors Associated With Disordered Eating Behavior Among Adolescent Girls: Screening and Education. J Prim Care Community Health. 2022;13:21501319211062673. doi:10.1177/21501319211062673.

Hedman L, Backman H, Stridsman C, Lundbäck M, Andersson M, Rönmark E. Predictors of electronic cigarette use among Swedish teenagers: a population-based cohort study. BMJ Open. 2020;10(12):e040683. doi:10.1136/bmjopen-2020-040683.

Wang M, Wang H, Hu RY, Gong WW, Pan J, Yu M. Associations between trying to control weight, weight control behaviors and current electronic cigarette usage in middle and high school students: A cross-sectional study in Zhejiang Province, China. Tob Induc Dis. 2020;18:28. doi:10.18332/tid/119126.

Ganson KT, Lavender JM, Rodgers RF, Cunningham M, Nagata JM. Compulsive exercise and vaping among a sample of U.S. College students aged 18-26 years. Eat Weight Disord. 2022;27(3):1153-1161. doi:10.1007/s40519-021-01251-z.

Liu Z, Zhang Y, Youn JY, Zhang Y, Makino A, Yuan JX, Cai H. Flavored and Nicotine-Containing E-Cigarettes Induce Impaired Angiogenesis and Diabetic Wound Healing via Increased Endothelial Oxidative Stress and Reduced NO Bioavailability. Antioxidants (Basel). 2022;11(5):904. doi:10.3390/antiox11050904.

Hod R, Mohd Nor NH, Maniam S. Systematic review on e-cigarette and its effects on weight gain and adipocytes. PLoS One. 2022;17(7):e0270818. doi:10.1371/journal.pone.0270818.

Tsai M, Byun MK, Shin J, Crotty Alexander LE. Effects of e-cigarettes and vaping devices on cardiac and pulmonary physiology. J Physiol. 2020;598(22):5039-5062. doi:10.1113/JP279754.

Aranyosi JK, Galgoczi E, Erdei A, Katko M, Fodor M, Ujhelyi Z, Bacskay I, Nagy EV, Ujhelyi B. Different Effects of Cigarette Smoke, Heated Tobacco Product and E-Cigarette Vapour on Orbital Fibroblasts in Graves' Orbitopathy; a Study by Real Time Cell Electronic Sensing. Molecules. 2022;27(9):3001. doi:10.3390/molecules27093001.

Górna I, Napierala M, Florek E. Electronic Cigarette Use and Metabolic Syndrome Development: A Critical Review. Toxics. 2020;8(4):105. doi:10.3390/toxics8040105

Fiani B, Noblett C, Nanney JM, Gautam N, Pennington E, Doan T, Nikolaidis D. The Impact of "Vaping" Electronic Cigarettes on Spine Health. Cureus. 2020;12(6):e8907. doi:10.7759/cureus.8907

Reumann MK, Schaefer J, Titz B, Aspera-Werz RH, Wong ET, Szostak J, Häussling V, Ehnert S, Leroy P, Tan WT, Kuczaj A, Audretsch C, Springer F, Badke A, Augat P, Quentanilla-Fend L, Martella M, Lee KM, Peitsch MC, Hoeng J, Nussler AK. E-vapor aerosols do not compromise bone integrity relative to cigarette smoke after 6-month inhalation in an ApoE-/- mouse model. Arch Toxicol. 2020;94(6):2163-2177. doi:10.1007/s00204-020-02769-4.

Bhattacharya B, Narain V, Bondesson M. E-cigarette vaping liquids and the flavoring chemical cinnamaldehyde perturb bone, cartilage and vascular development in zebrafish embryos. Aquat Toxicol. 2021;240:105995. doi:10.1016/j.aquatox.2021.105995.

Nogueira L, Zemljic-Harpf AE, Yusufi R, Ranjbar M, Susanto C, Tang K, Mahata SK, Jennings PA, Breen EC. E-cigarette aerosol impairs male mouse skeletal muscle force development and prevents recovery from injury. Am J Physiol Regul Integr Comp Physiol. 2022;323(6):R849-R860. doi:10.1152/ajpregu.00314.2021.

Nicholson T, Davis L, Davis ET, Newton Ede M, Scott A, Jones SW. e-Cigarette Vapour Condensate Reduces Viability and Impairs Function of Human Osteoblasts, in Part, via a Nicotine Dependent Mechanism. Toxics. 2022;10(9):506. doi:10.3390/toxics10090506.

Micallef J, Batisse A, Revol B. Pharmacology of cannabidiol: Red flags, consequences and risks in humans. Therapie. 2022;77(5):585-590. doi:10.1016/j.therap.2022.02.001.

Menshov VA, Trofimov AV, Zagurskaya AV, Berdnikova NG, Yablonskaya OI, Platonova AG. Influence of Nicotine from Diverse Delivery Tools on the Autonomic Nervous and Hormonal Systems. Biomedicines. 2022;10(1):121. doi:10.3390/biomedicines10010121.

Benowitz NL, St Helen G, Nardone N, Addo N, Zhang JJ, Harvanko AM, Calfee CS, Jacob P 3rd. Twenty-Four-Hour Cardiovascular Effects of Electronic Cigarettes Compared With Cigarette Smoking in Dual Users. J Am Heart Assoc. 2020;9(23):e017317. doi:10.1161/JAHA.120.017317.

Jones CA, Wallace MJ, Bandaru P, Woodbury ED, Mohler PJ, Wold LE. E-cigarettes and arrhythmogenesis: a comprehensive review of preclinical studies and their clinical implications. Cardiovasc Res. 2023:cvad113. doi:10.1093/cvr/cvad113.

Mull ES, Erdem G, Nicol K, Adler B, Shell R. Eosinophilic Pneumonia and Lymphadenopathy Associated With Vaping and Tetrahydrocannabinol Use. Pediatrics. 2020;145(4):e20193007. doi:10.1542/peds.2019-3007.

Furlow B. US CDC issues guidance on e-cigarette, or vaping, associated lung injury. Lancet Respir Med. 2020;8(1):20. doi:10.1016/S2213-2600(19)30409-6.

April-Sanders AK, Daviglus ML, Lee UJ, Perreira KM, Kaplan RC, Blaha MJ, Pirzada A, Giachello AL, Bhatnagar A, Robertson RM, Thanh-Huyen TV, Rodriguez CJ. Prevalence of electronic cigarette use and its determinants in us persons of Hispanic/Latino background: The Hispanic community health study / study of Latinos (HCHS/SOL). Am J Med Open. 2023 Jun;9:100029. doi: 10.1016/j.ajmo.2022.100029.

Chelikam N, Mohammad Z, Tavrawala K, Krishnakumar AN, Varghese A, Shrivastav TY, Tarimci B, Kumar S, Francis SZ, Samala Venkata V, Patel UK, Manjani L. Prevalence of Cerebrovascular Accidents Among the US Population With Substance Use Disorders: A Nationwide Study.Cureus. 2022 Nov 23;14(11):e31826. doi: 10.7759/cureus.31826

Richardson A, Krivokhizhina T, Lorkiewicz P, D'Souza S, Bhatnagar A, Srivastava S, Conklin DJ. Effects of electronic cigarette flavorants on human platelet aggregation ex vivo. Toxicol Rep. 2022 Apr 8;9:814-820. doi: 10.1016/j.toxrep.2022.04.003.

Dahdah A, Jaggers RM, Sreejit G, Johnson J, Kanuri B, Murphy AJ, Nagareddy PR. Immunological Insights into Cigarette Smoking-Induced Cardiovascular Disease Risk. Cells. 2022 Oct 11;11(20):3190. doi: 10.3390/cells11203190.

Scharf P, Rizzetto F, Xavier LF, Farsky SHP.
Xenobiotics Delivered by Electronic Nicotine Delivery Systems: Potential Cellular and Molecular Mechanisms on the Pathogenesis of Chronic Kidney Disease. Int J Mol Sci. 2022 Sep 7;23(18):10293. doi: 10.3390/ijms231810293.

Rodríguez-Bolaños R, Baruch-Dominguez R, Arillo-Santillán E, Yunes-Díaz E, Rivera-Rivera L, Cruz-Jiménez L, Thrasher JF, Nyitray AG, Lazcano-Ponce E. Latent Class Analysis of Polysubstance Use and Sexual Risk Behaviors among Men Who Have Sex with Men Attending Sexual Health Clinics in Mexico City. Int J Environ Res Public Health. 2022 Jul 21;19(14):8847. doi: 10.3390/ijerph19148847.

Burnett T, Battista K, Butt M, Sherifali D, Leatherdale ST, Dobbins M.The association between public health engagement in school-based substance use prevention programs and student alcohol, cannabis, e-cigarette and cigarette use.
Can J Public Health. 2023 Feb;114(1):94-103. doi: 10.17269/s41997-022-00655-3

Assi HI, Meouchy P, El Mahmoud A, Massouh A, Bou Zerdan M, Alameh I, Chamseddine N, Kazarian H, Zeineldine S, Saliba NA, Noureddine S.A Survey on the Knowledge, Attitudes, and Practices of Lebanese Physicians Regarding Air Pollution.
Int J Environ Res Public Health. 2022 Jun 28;19(13):7907. doi: 10.3390/ijerph19137907.

Ren X, Lin L, Sun Q, Li T, Sun M, Sun Z, Duan J.Metabolomics-based safety evaluation of acute exposure to electronic cigarettes in mice.
Sci Total Environ. 2022 Sep 15;839:156392. doi: 10.1016/j.scitotenv.2022.156392.

AlMulla A, Mamtani R, Cheema S, Maisonneuve P, Daghfal J, Kouyoumjian S. The association between tobacco use and COVID-19 in Qatar.Prev Med Rep. 2022 Aug;28:101832. doi: 10.1016/j.pmedr.2022.101832.

AlQobaly L, Abed H, Alsahafi Y, Sabbah W, Hakeem FF. Does smoking explain the association between use of e-cigarettes and self-reported periodontal disease?

J Dent. 2022 Jul;122:104164. doi: 10.1016/j.jdent.2022.104164.

Alharthy A, Hosler AS, Leckman-Westin E, Kammer JR.Association Between Tobacco Retailer Density and Smoking Among Adults With Diabetes and Serious Mental Illness in New York State. Prev Chronic Dis. 2022 Jan 6;19:E01. doi: 10.5888/pcd19.210270.

Chin A, Zonfrillo MR, Heinly A, Ford SR, Quintos JB, Topor LS.Screening and counseling for nicotine use in youth with diabetes.
Pediatr Diabetes. 2022 Feb;23(1):157-164. doi: 10.1111/pedi.13287. Epub 2021 Dec 12.

Sayed IM, Masso-Silva JA, Mittal A, Patel A, Lin E, Moshensky A, Shin J, Bojanowski CM, Das S, Akuthota P, Crotty Alexander LEInflammatory phenotype modulation in the respiratory tract and systemic circulation of e-cigarette users: a pilot study.  Am J Physiol Lung Cell Mol Physiol. 2021 Dec 1;321(6):L1134-L1146.

Li W, Pan J, Wei M, Lv Z, Chen S, Qin Y, Li N. Nonocular Influencing Factors for Primary Glaucoma: An Umbrella Review of Meta-Analysis. Ophthalmic Res. 2021;64(6):938

Bricknell RAT, Ducaud C, Figueroa A, Schwarzman LS, Rodriguez P, Castro G, Zevallos JC, Barengo NC.An association between electronic nicotine delivery systems use and a history of stroke using the 2016 behavioral risk factor surveillance system.Medicine (Baltimore). 2021 Sep 10;100(36):e27180. doi: 10.1097/MD.0000000000027180.

Bush A, Ferkol T, Valiulis A, Mazur A, Chkhaidze I, Maglakelidze T, Sargsyan S, Boyajyan G, Cirstea O, Doan S, Katilov O, Pokhylko V, Dubey L, Poluziorovienė E, Prokopčiuk N, Taminskienė V, Valiulis A.Unfriendly Fire: How the Tobacco Industry is Destroying the Future of Our Children.
Acta Med Litu. 2021;28(1):6-18. doi: 10.15388/Amed.2020.28.1.6.

Daou MAZ, Shihadeh A, Hashem Y, Bitar H, Kassir A, El-Harakeh M, Karaoghlanian N, Eid AA, El-Sabban M, Zaatari G, Husari A.Role of diabetes in lung injury from acute exposure to electronic cigarette, heated tobacco product, and combustible cigarette aerosols in an animal model. PLoS One. 2021 Aug 10;16(8):e0255876. doi: 10.1371/journal.pone.0255876.

Shah C, Singh P, Matin S, Farrow J, Magon R, Zia A, Tatt-Smith P, Watson C, Smith AA physician associate-led clinic for people with severe mental illness in the United Kingdom. JAAPA. 2021 Aug 1;34(8):1-6. doi: 10.1097/01.JAA.0000758220.38067.49.

Marques P, Piqueras L, Sanz MJ. An updated overview of e-cigarette impact on human health. Respir Res. 2021 May 18;22(1):151. doi: 10.1186/s12931-021-01737-5.

Herout KT, Durant EJ, Fong J. Dysphagia as the Predominant Symptom in Posterior Circulation Stroke: A Case Report. Am J Case Rep. 2021 May 18;22:e930502.

Liu J, Niederdeppe J. Misperceptions of the Prevalence of Health Conditions and Behaviors. J Health Commun. 2020 Nov 1;25(11):903-916. doi: 10.1080/10810730.2020.1858461.

Kos K. Cardiometabolic Morbidity and Mortality with Smoking Cessation, Review of Recommendations for People with Diabetes and Obesity. Curr Diab Rep. 2020 Dec 8;20(12):82. doi: 10.1007/s11892-020-01352-6.

Atuegwu NC, Oncken C, Laubenbacher RC, Perez MF, Mortensen EM.Factors Associated with E-Cigarette Use in U.S. Young Adult Never Smokers of Conventional Cigarettes: A Machine Learning Approach. Int J Environ Res Public Health. 2020 Oct 5;17(19):7271. doi: 10.3390/ijerph17197271.

Lynch J, Jin L, Richardson A, Conklin DJ. Tobacco Smoke and Endothelial Dysfunction: Role of Aldehydes? Curr Hypertens Rep. 2020 Aug 28;22(9):73. doi: 10.1007/s11906-020-01085-7.

Lan K, Zhang G, Liu L, Guo Z, Luo X, Guan H, Yu Q, Liu E. Electronic cigarette exposure on insulin sensitivity of ApoE gene knockout mice. Tob Induc Dis. 2020 Aug 6;18:68. doi: 10.18332/tid/125399. eCollection 2020.

Smith LC, George O. Advances in smoking cessation pharmacotherapy: Non-nicotinic approaches in animal models. Neuropharmacology. 2020 Nov 1;178:108225.

Balogh E, Wagner Z, Faubl N, Riemenschneider H, Voigt K, Terebessy A, Horváth F, Füzesi Z, Kiss I. Increasing Prevalence of Electronic Cigarette Use among Medical Students. Repeated Cross-Sectional Multicenter Surveys in Germany and Hungary, 2016-2018. Subst Use Misuse. 2020;55(13):2109-2115.

Rutledge KJ, Plath DL. Acute Psychosis in Withdrawal from Nicotine Vaping in a Young Man with Comorbid Diabetic Ketoacidosis and Cannabis Use. Case Rep Psychiatry. 2020 Jun 3;2020:5710810.

Mattingly DT, Pfeiffer J, Tompkins LK, Rai J, Sears CG, Walker KL, Hart JL. Associations between Appalachian youth tobacco consumption and communication channel use.Tob Prev Cessat. 2020 Mar 24;6:21.

Debarba LK, Mulka A, Lima JBM, Didyuk O, Fakhoury P, Koshko L, Awada AA, Zhang K, Klueh U, Sadagurski M.Acarbose protects from central and peripheral metabolic imbalance induced by benzene exposure. Brain Behav Immun. 2020 Oct;89:87-99.

Chaffee BW, Persai D, Vora MV. Interdental Cleaning and Oral Health Status in an Adult Cohort, 2015 to 2018. J Dent Res. 2020 Sep;99(10):1150-1156.

AJPH Global News. [No authors listed] Am J Public Health. 2020 Jun;110(6):751. doi: 10.2105/AJPH.2020.305665.

Parekh T, Pemmasani S, Desai R. Marijuana Use Among Young Adults (18-44 Years of Age) and Risk of Stroke: A Behavioral Risk Factor Surveillance System Survey Analysis. Stroke. 2020 Jan;51(1):308-310.

Ali D, Kuyunov I, Baskaradoss JK, Mikami T. Comparison of periodontal status and salivary IL-15 and -18 levels in cigarette-smokers and individuals using electronic nicotine delivery systems.BMC Oral Health. 2022 Dec 30;22(1):655.

Silveira ML, Everard CD, Sharma E, Lauten K, Alexandridis AA, Duffy K, Taylor EV, Tolliver EA, Blanco C, Compton WM, Kimmel HL, Iafolla T, Hyland A, Chaffee BW.
Tobacco Use and Incidence of Adverse Oral Health Outcomes Among US Adults in the Population Assessment of Tobacco and Health Study. JAMA Netw Open. 2022 Dec 1;5(12):e2245909

Youssef M, Marzouk T, Abdelsalam H, Malmstrom H, Barmak AB, Fraser D, Tsigarida A. The effect of electronic cigarette use on peri-implant conditions in men: a systematic review and meta-analysis. Oral Surg Oral Med Oral Pathol Oral Radiol. 2023 Apr;135(4):492-500.

Ruedisueli I, Arastoo S, Gupta PK, Gornbein J, Middlekauff HR. Neural-hematopoietic-inflammatory axis in nonsmokers, electronic cigarette users, and tobacco smokers. Physiol Rep. 2022 Oct;10(19):e15412. doi: 10.14814/phy2.15412.

Alhumaidan AA, Al-Aali KA, Vohra F, Javed F, Abduljabbar T. Comparison of Whole Salivary Cortisol and Interleukin 1-Beta Levels in Light Cigarette-Smokers and Users of Electronic Nicotine Delivery Systems before and after Non-Surgical Periodontal Therapy. Int J Environ Res Public Health. 2022 Sep 8;19(18):11290.

Thomas SC, Xu F, Pushalkar S, Lin Z, Thakor N, Vardhan M, Flaminio Z, Khodadadi-Jamayran A, Vasconcelos R, Akapo A, Queiroz E, Bederoff M, Janal MN, Guo Y, Aguallo D, Gordon T, Corby PM, Kamer AR, Li X, Saxena D. Electronic Cigarette Use Promotes a Unique Periodontal Microbiome. mBio. 2022 Feb 22;13(1):e0007522.

Armstrong ML, Smith N, Tracey R, Jackman H. The Orthopedic Effects of Electronic Cigarettes: A Systematic Review and Pediatric Case Series. Children (Basel). 2022 Jan 4;9(1):62

Alazmi SO, Almutairi FJ, Alresheedi BA. Comparison of Peri-Implant Clinicoradiographic Parameters among Non-Smokers and Individuals Using Electronic Nicotine Delivery Systems at 8 Years of Follow-up. Oral Health Prev Dent. 2021 Jan 7;19(1):511-516.

Dekhou A, Oska N, Partiali B, Johnson J, Chung MT, Folbe A. E-Cigarette Burns and Explosions: What are the Patterns of Oromaxillofacial Injury? J Oral Maxillofac Surg. 2021 Aug;79(8):1723-1730.

Morris TM, Marlborough FJ, Montgomery RJ, Allison KP, Eardley WGP. Smoking and the patient with a complex lower limb injury. Injury. 2021 Apr;52(4):814-824.

Figueredo CA, Abdelhay N, Figueredo CM, Catunda R, Gibson MP. The impact of vaping on periodontitis: A systematic review. Clin Exp Dent Res. 2021 Jun;7(3):376-384

Rogers SW, Myers EJ, Gahring LC. Age-Associated Tooth Loss and Oral Microbial Dysbiosis in a Mouse Genetic Model of Chronic Nicotine Exposure.Front Immunol. 2020 Oct 7;11:575200

Derespina KR, Kaushik S, Mitchell W, Gorstein S, Ushay HM, Medar SS. E-cigarette or Vaping-Associated Acute Lung Injury and Hemophagocytic Lymphohistiocytosis. Pediatrics. 2020 Oct;146(4):e20193664.

Aspera-Werz RH, Ehnert S, Müller M, Zhu S, Chen T, Weng W, Jacoby J, Nussler AK. Assessment of tobacco heating system 2.4 on osteogenic differentiation of mesenchymal stem cells and primary human osteoblasts compared to conventional cigarettes. World J Stem Cells. 2020 Aug 26;12(8):841-856.

Ramanathan G, Craver-Hoover B, Arechavala RJ, Herman DA, Chen JH, Lai HY, Renusch SR, Kleinman MT, Fleischman AG. E-Cigarette Exposure Decreases Bone Marrow Hematopoietic Progenitor Cells. Cancers (Basel). 2020 Aug 14;12(8):2292.

Fiani B, Noblett C, Nanney JM, Gautam N, Pennington E, Doan T, Nikolaidis D. The Impact of "Vaping" Electronic Cigarettes on Spine Health. Cureus. 2020 Jun 29;12

Vyncke T, De Wolf E, Hoeksema H, Verbelen J, et al. Injuries associated with electronic nicotine delivery systems: A systematic review. J Trauma Acute Car. Surg.2020;89(4):783-791.

Aldakheel FM, Alduraywish SA, Jhugroo P, Jhugroo C, Divakar DD. Quantification of pathogenic bacteria in the subgingival oral biofilm samples collected from cigarette-smokers, individuals using electronic nicotine delivery systems and non-smokers with and without periodontitis.Arch Oral Biol. 2020 Sep;117:104793.

Chaffee BW, Persai D, Vora MV. Interdental Cleaning and Oral Health Status in an Adult Cohort, 2015 to 2018. J Dent Res. 2020 Sep;99(10):1150-1156.

Ibraheem WI, Fageeh HI, Preethanath RS, Alzahrani FA, Al-Zawawi AS, Divakar DD, Al-Kheraif AA. Comparison of RANKL and osteoprotegerin levels in the gingival crevicular fluid of young cigarette- and waterpipe-smokers and individuals using electronic nicotine delivery systems. Arch Oral Biol. 2020 Jul;115:104714.

Al Deeb M, Alresayes S, A Mokeem S, Alhenaki AM, AlHelal A, Vohra F, Abduljabbar T. Clinical peri-implant health and biological bone marker levels in tobacco users treated with photodynamic therapy. Photodiagnosis Photodyn Ther. 2020 Sep;31:101821.

Al-Hamoudi N, Alsahhaf A, Al Deeb M, Alrabiah M, Vohra F, Abduljabbar T. Effect of scaling and root planing on the expression of anti-inflammatory cytokines (IL-4, IL-9, IL-10, and IL-13) in the gingival crevicular fluid of electronic cigarette users and non-smokers with moderate chronic periodontitis. J Periodontal Implant Sci. 2020 Feb 19;50(2):74-82.

Vohra F, Bukhari IA, Sheikh SA, Albaijan R, Naseem M. Comparison of self-rated oral symptoms and periodontal status among cigarette smokers and individuals using electronic nicotine delivery systems. J Am Coll Health. 2020 Oct;68(7):788-793.

Reddy V, Wurtz M, Patel SH, McCarthy M, Raval AP. Oral contraceptives and stroke: Foes or friends. Front Neuroendocrinol. 2022 Oct;67:101016

Chin A, Zonfrillo MR, Heinly A, Ford SR, Quintos JB, Topor LS. Screening and counseling for nicotine use in youth with diabetes. Pediatr Diabetes. 2022 Feb;23(1):157-164.

Wang L, Du L, Xiong X, Lin Y, Zhu J, Yao Z, Wang S, Guo Y, Chen Y, Geary K, Pan Y, Zhou F, Gao S, Zhang D, Yeung SJ, Zhang H. Repurposing dextromethorphan and metformin for treating nicotine-induced cancer by directly targeting CHRNA7 to inhibit JAK2/STAT3/SOX2 signaling. Oncogene. 2021 Mar;40(11):1974-1987.

Casagrande M, Favieri F, Guarino A, Di Pace E, Langher V, Germanò G, Forte G. The Night Effect of Anger: Relationship with Nocturnal Blood Pressure Dipping. Int J Environ Res Public Health. 2020 Apr 15;17(8):2705.

Reidel B, Abdelwahab S, Wrennall JA, Clapp PW, Beers JL, Jackson KD, Tarran R, Kesimer M. Vaping additives cannabinoid oil and vitamin E acetate adhere to and damage the human airway epithelium. J Appl Toxicol. 2023 May;43(5):680-693.

Rose JJ, Rebuli ME, Noël A, Croft DP. Clearing Some of the Haze around E-cigarette or Vaping Product Use-Associated Lung Injury (EVALI). Ann Am Thorac Soc. 2022 Nov;19(11):1805-1807.

Geci M, Scialdone M, Tishler J. The Dark Side of Cannabidiol: The Unanticipated Social and Clinical Implications of Synthetic Δ8-THC. Cannabis Cannabinoid Res. 2023 Apr;8(2):270-282.

Boakye E, El Shahawy O, Obisesan O, Dzaye O, Osei AD, Erhabor J, Uddin SMI, Blaha MJ. The inverse association of state cannabis vaping prevalence with the e-cigarette or vaping product-use associated lung injury. PLoS One. 2022 Oct 17;17(10):e0276187.

Fried ND, Whitehead A, Lazartigues E, Yue X, Gardner JD. Ovarian hormones do not mediate protection against pulmonary hypertension and right ventricular remodeling in female mice exposed to chronic, inhaled nicotine. Am J Physiol Heart Circ Physiol. 2022 Nov 1;323(5):H941-H948.

Katchmar A, Shafer P, Siegel M. Analysis of state portrayals of the risks of e-cigarette use and the cause of the EVALI outbreak. Harm Reduct J. 2022 Oct 5;19(1):112.

Hindelang P, Scharinger A, Golombek P, Laible M, Tamosaite S, Walch SG, Lachenmeier DW. Absence of Relevant Thermal Conversion of Cannabidiol to Tetrahydrocannabinol in E-Cigarette Vapor and Low-Tetrahydrocannabinol Cannabis Smoke. Cannabis Cannabinoid Res. 2022 Oct;7(5):616-620.

La Maida N, Di Giorgi A, Pichini S, Busardò FP, Huestis MA. Recent challenges and trends in forensic analysis: Δ9-THC isomers pharmacology, toxicology and analysis. J Pharm Biomed Anal. 2022 Oct 25;220:114987

Berg CJ, Melena A, Wittman FD, Robles T, Henriksen L. The Reshaping of the E-Cigarette Retail Environment: Its Evolution and Public Health Concerns. Int J Environ Res Public Health. 2022 Jul 12;19(14):8518. 

Sun R, Mendez D, Warner KE. Use of Electronic Cigarettes Among Cannabis-Naive Adolescents and Its Association With Future Cannabis Use. JAMA Netw Open. 2022 Jul 1;5(7):e2223277

Marrocco A, Singh D, Christiani DC, Demokritou P. E-cigarette vaping associated acute lung injury (EVALI): state of science and future research needs.
 Crit Rev Toxicol. 2022 Mar;52(3):188-220.

Breit KR, Rodriguez CG, Lei A, Hussain S, Thomas JD. Effects of prenatal alcohol and delta-9-tetrahydrocannabinol exposure via electronic cigarettes on motor development.Alcohol Clin Exp Res. 2022 Aug;46(8):1408-1422

Frinculescu A, Coombes G, Shine T, Ramsey J, Johnston A, Couchman L. Analysis of illicit drugs in purchased and seized electronic cigarette liquids from the United Kingdom 2014-2021. Drug Test Anal. 2023 Oct;15(10):1058-1066.

Manandhar A, Haron MH, Ross SA, Klein ML, Elokely KM. Potential Pro-Inflammatory Effect of Vitamin E Analogs through Mitigation of Tetrahydrocannabinol (THC) Binding to the Cannabinoid 2 Receptor. Int J Mol Sci. 2022 Apr 13;23(8):4291.

Davis DR, Bold KW, Kong G, Cavallo DA, Jackson A, Krishnan-Sarin S. Cannabis use among youth who vape nicotine E-cigarettes: A qualitative analysis. Drug Alcohol Depend. 2022 May 1;234:109413.

McGraw MD, Croft DP, Nacca NE, Rahman I. Reduced plasma phosphatidylethanolamines in e-cigarette, or vaping, product use-associated lung injury (EVALI). Pediatr Pulmonol. 2022 May;57(5):1350-1354.

Wackowski OA, Gratale SK, Jeong M, Delnevo CD, Steinberg MB, O'Connor RJ. Over 1 year later: smokers' EVALI awareness, knowledge and perceived impact on e-cigarette interest. Tob Control. 2023 Aug;32(e2):e255-e259.

Micallef J, Batisse A, Revol B. [Pharmacology of cannabidiol: Red flags, consequences and risks in humans]. Therapie. 2022 Sep-Oct;77(5):585-590.

Parks MJ, Maggs JL, Patrick ME. Daily fluctuations in drinking intensity: Links with vaping and combustible use of nicotine and marijuana. Drug Alcohol Depend. 2022 Apr 1;233:109347

LeBouf RF, Ranpara A, Ham J, Aldridge M, Fernandez E, Williams K, Burns DA, Stefaniak AB. Chemical Emissions From Heated Vitamin E Acetate-Insights to Respiratory Risks From Electronic Cigarette Liquid Oil Diluents Used in the Aerosolization of Δ9-THC-Containing Products. Front Public Health. 2022 Jan 21;9:765168

Lallai V, Manca L, Sherafat Y, Fowler CD. Effects of Prenatal Nicotine, THC, or Co-Exposure on Cognitive Behaviors in Adolescent Male and Female Rats. Nicotine Tob Res. 2022 Jul 13;24(8):1150-1160.

Baker MM, Procter TD, Belzak L, Ogunnaike-Cooke S. Vaping-associated lung illness (VALI) in Canada: a descriptive analysis of VALI cases reported from September 2019 to December 2020. Health Promot Chronic Dis Prev Can. 2022 Jan;42(1):37-44.

Joglekar R, Cauley M, Lipsich T, Corcoran DL, Patisaul HB, Levin ED, Meyer JN, McCarthy MM, Murphy SK. Developmental nicotine exposure and masculinization of the rat preoptic area. Neurotoxicology. 2022 Mar;89:41-54

Theophilopoulos JM, LeLaurin JH, Williams M, Bright M, Thompson LA, Salloum RG. Provider documentation of electronic nicotine delivery systems use among patients prescribed contraception at an academic health center in the Southeastern United States. Prev Med Rep. 2021 Nov 11;24:101632

Wang Y, Tong YQ, Zhou W, Tian ZL, Li NN, Lyu XX, Sun TY, Ke HX. [Electronic cigarette use-associated lung injury: a case report and literature review]. Zhonghua Jie He He Hu Xi Za Zhi. 2021 May 12;44(5):481-487.

Ranpara A, Stefaniak AB, Williams K, Fernandez E, LeBouf RF. Modeled Respiratory Tract Deposition of Aerosolized Oil Diluents Used in Δ9-THC-Based Electronic Cigarette Liquid Products. Front Public Health. 2021 Nov 4;9:744166.

Al-Sawalha NA, Bdeir R, Sohaib A, Saad M, Inghaimesh T, Khabour OF, Alzoubi KH, Shihadeh A. Effect of E-cigarettes aerosol exposure during lactation in rats: Hormonal and biochemical aspects. Environ Toxicol Pharmacol. 2021 Nov;88:103759

Kovach AL, Carter RR, Thornburg JW, Wiethe R, Fennell TR, Wiley JL. Thermal Degradants Identified from the Vaping of Vitamin E Acetate. J Anal Toxicol. 2022 Aug 13;46(7):750-756

Duffy BC, Li L, Lu S, Dittmar MA, Delaney-Baldwin E, Durocher LA, Spink DC. Chemotyping of ∆8-THC-Containing e-Liquids Analyzed during the 2019-2020 New York State EVALI Investigation. J Anal Toxicol. 2022 Aug 13;46(7):743-749.

Meehan-Atrash J, Rahman I. Cannabis Vaping: Existing and Emerging Modalities, Chemistry, and Pulmonary Toxicology. Chem Res Toxicol. 2021 Oct 18;34(10):2169-2179

Ruiz CM, Torrens A, Lallai V, Castillo E, Manca L, Martinez MX, Justeson DN, Fowler CD, Piomelli D, Mahler SV. Pharmacokinetic and pharmacodynamic properties of aerosolized ("vaped") THC in adolescent male and female rats. Psychopharmacology (Berl). 2021 Dec;238(12):3595-3605.

Chan BS, Kiss A, McIntosh N, Sheppeard V, Dawson AH. E-cigarette or vaping product use-associated lung injury in an adolescent. Med J Aust. 2021 Oct 4;215(7):313-314.e1.

Gutierrez A, Creehan KM, Turner ML, Tran RN, Kerr TM, Nguyen JD, Taffe MA. Vapor exposure to Δ9-tetrahydrocannabinol (THC) slows locomotion of the Maine lobster (Homarus americanus). Pharmacol Biochem Behav. 2021 Aug;207:173222.

Forbes TP, Krauss ST. Confined DART-MS for Rapid Chemical Analysis of Electronic Cigarette Aerosols and Spiked Drugs. J Am Soc Mass Spectrom. 2021 Aug 4;32(8):2274-2280.

Gutierrez A, Nguyen JD, Creehan KM, Javadi-Paydar M, Grant Y, Taffe MA. Effects of combined THC and heroin vapor inhalation in rats. Psychopharmacology (Berl). 2022 May;239(5):1321-1335.

Trivers KF, Watson CV, Neff LJ, Jones CM, Hacker K. Tetrahydrocannabinol (THC)-containing e-cigarette, or vaping, product use behaviors among adults after the onset of the 2019 outbreak of e-cigarette, or vaping, product use-associated lung injury (EVALI). Addict Behav. 2021 Oct;121:106990.

Turna J, Balodis I, Van Ameringen M, Busse JW, MacKillop J. Attitudes and Beliefs Toward Cannabis Before Recreational Legalization: A Cross-Sectional Study of Community Adults in Ontario.Cannabis Cannabinoid Res. 2022 Aug;7(4):526-536.

 Kligerman SJ, Kay FU, Raptis CA, Henry TS, Sechrist JW, Walker CM, Vargas D, Filev PD, Chung MS, Digumarthy SR, Ropp AM, Mohammed TL, Pope KW, Marquis KM, Chung JH, Kanne JP. CT Findings and Patterns of e-Cigarette or Vaping Product Use-Associated Lung Injury: A Multicenter Cohort of 160 Cases.
 Chest. 2021 Oct;160(4):1492-1511.

Kaslow JA, Rosas-Salazar C, Moore PE. E-cigarette and vaping product use-associated lung injury in the pediatric population: A critical review of the current literature. Pediatr Pulmonol. 2021 Jul;56(7):1857-1867.

Czégény Z, Nagy G, Babinszki B, Bajtel Á, Sebestyén Z, Kiss T, Csupor-Löffler B, Tóth B, Csupor D. CBD, a precursor of THC in e-cigarettes. Sci Rep. 2021 Apr 26;11(1):8951.

Renda B, Andrade AK, Stone APS, El Azali R, Sharivker M, Khokhar JY, Antenos M, Murray JE. Adolescent nicotine and footshock exposure augments adult nicotine self-administration and drug-seeking without affecting baseline anxiety-like behaviour or stress responsivity in male rats. Psychopharmacology (Berl). 2021 Jun;238(6):1687-1701.

Algiers O, Wang Y, Laestadius L. Content Analysis of U.S. Newspaper Coverage of Causes and Solutions to Vaping-Associated Lung Injury. Subst Use Misuse. 2021;56(4):522-528

Lucero A, Eriksson N, Nichta C, Sokol K. A 23-year-old man with acute lung injury after using a tetrahydrocannabinol-containing vaping device: a case report. J Med Case Rep. 2021 Feb 11;15(1):70.

Casula L, Sinico C, Valenti D, Pini E, Pireddu R, Schlich M, Lai F, Maria Fadda A. Delivery of beclomethasone dipropionate nanosuspensions with an electronic cigarette. Int J Pharm. 2021 Mar 1;596:120293

Carboni L, Ponzoni L, Braida D, Sala M, Gotti C, Zoli M. Altered mRNA Levels of Stress-Related Peptides in Mouse Hippocampus and Caudate-Putamen in Withdrawal after Long-Term Intermittent Exposure to Tobacco Smoke or Electronic Cigarette Vapour. Int J Mol Sci. 2021 Jan 9;22(2):599

Mado H, Reichman-Warmusz E, Wojnicz R. The vaping product use associated lung injury: is this a new pulmonary disease entity? Rev Environ Health. 2020 Dec 7;36(2):145-157.

Arons MM, Barnes SR, Cheng R, Whittle K, Elsholz C, Bui D, Gilley S, Maldonado A, LaCross N, Sage K, Lewis N, McCaffrey K, Green J, Duncan J, Dunn AC. Examining the temporality of vitamin E acetate in illicit THC-containing e-cigarette, or vaping, products from a public health and law enforcement response to EVALI - Utah, 2018-2020. Int J Drug Policy. 2021 Feb;88:103026.

Cwalina SN, Braymiller JL, Leventhal AM, Unger JB, McConnell R, Barrington-Trimis JL. Prevalence of Young Adult Vaping, Substance Vaped, and Purchase Location Across Five Categories of Vaping Devices. Nicotine Tob Res. 2021 May 4;23(5):829-835.

Wagner J, Chen W, Vrdoljak G. Vaping cartridge heating element compositions and evidence of high temperatures. PLoS One. 2020 Oct 19;15(10):e0240613

Choi H, Lin Y, Race E, Macmurdo MG. Electronic Cigarettes and Alternative Methods of Vaping. Ann Am Thorac Soc. 2021 Feb;18(2):191-199.

Espinosa SM, Harper EP, Phillips MB. 19-Year-Old Man With Fevers, Abdominal Pain, and Cough. Mayo Clin Proc. 2020 Oct;95(10):e103-e108.

 Kessler AC, Dommann C, Nussbaumer-Ochsner Y. E-pipe use leading to lipoid pneumonia in Europe. Thorax. 2020 Nov;75(11):1026-1027.

Kleinman MT, Arechavala RJ, Herman D, Shi J, Hasen I, Ting A, Dai W, Carreno J, Chavez J, Zhao L, Kloner RA.  E-cigarette or Vaping Product Use-Associated Lung Injury Produced in an Animal Model From Electronic Cigarette Vapor Exposure Without Tetrahydrocannabinol or Vitamin E Oil. J Am Heart Assoc. 2020 Sep 15;9(18):e017368.

Maslonka MA, Schertz AR, Markowski LM, Miller PJ. Sedation challenges in patients with E-cigarette, or vaping, product use-associated lung injury (EVALI). BMJ Case Rep. 2020 Sep 2;13(9):e233866

Navon L, Ghinai I, Layden J. Notes from the Field: Characteristics of Tetrahydrocannabinol-Containing E-cigarette, or Vaping, Products Used by Adults - Illinois, September-October 2019. MMWR Morb Mortal Wkly Rep. 2020 Jul 24;69(29):973-975.

Striley CW, Nutley SK. World vaping update. Curr Opin Psychiatry. 2020 Jul;33(4):360-368.

Hage R, Fretz V, Schuurmans MM. Electronic cigarettes and vaping associated pulmonary illness (VAPI): A narrative review. Pulmonology. 2020 Sep-Oct;26(5):291-303.

Odish MF, Bellinghausen A, Golts E, Owens RL. E-cigarette, or vaping, product use-associated lung injury (EVALI) treated with veno-venous extracorporeal membrane oxygenation (VV-ECMO) and ultra-protective ventilator settings. BMJ Case Rep. 2020 Jul 2;13(7):e234771.

Poschenrieder F, Rotter M, Gschwendtner A, Hamer OW.E-cigarette-induced lung disease: from acute to chronic. Lancet. 2020 Aug 22;396(10250):564.

Brighenti V, Protti M, Anceschi L, Zanardi C, Mercolini L, Pellati F. Emerging challenges in the extraction, analysis and bioanalysis of cannabidiol and related compounds. J Pharm Biomed Anal. 2021 Jan 5;192:113633.

Print Screen

Round 4

Reviewer 1 Report

Comments and Suggestions for Authors

The authors have addressed any revisions I previously made note of.